# Aging with GRACE: Lifelong Model Editing with Discrete Key-Value Adaptors

**Thomas Hartvigsen**
University of Virginia, MIT
hartvigsen@virginia.edu

**Swami Sankaranarayanan**
Sony AI
swami.sankaranarayanan@sony.com

**Hamid Palangi**
Microsoft Research
hpalangi@microsoft.com

**Yoon Kim**
MIT
yoonkim@mit.edu

**Marzyeh Ghassemi**
MIT
mghassem@mit.edu

## Abstract

Deployed language models decay over time due to shifting inputs, changing user needs, or emergent world-knowledge gaps. When such problems are identified, we want to make targeted edits while avoiding expensive retraining. However, current model editors, which modify such behaviors of pre-trained models, degrade model performance quickly across multiple, sequential edits. We propose GRACE, a *lifelong* model editing method, which implements spot-fixes on streaming errors of a deployed model, ensuring minimal impact on unrelated inputs. GRACE writes new mappings into a pre-trained model's latent space, creating a discrete, local codebook of edits without altering model weights. This is the first method enabling thousands of sequential edits using only streaming errors. Our experiments on T5, BERT, and GPT models show GRACE's state-of-the-art performance in making and retaining edits, while generalizing to unseen inputs. Our code is available at github.com/thartvigsen/grace.

## 1 Introduction

Large scale pre-trained neural networks are the state-of-the-art for many hard machine learning problems, especially in natural language processing [5, 33] and computer vision [9, 34]. But when deployed, they still make unpredictable errors [2, 43]. For example, Large Language Models (LLMs) notoriously *hallucinate* [17], *perpetuate bias* [11], and *factually decay* [8]. Many such errors will arise sequentially in deployment, some of which must be addressed quickly without waiting until new training data is collected [16, 21]. For example, when an LLM generates hate speech or crucial knowledge about the world changes, its behavior must be modified immediately to protect users.

While *retraining* or *finetuning* can edit a model's predictions, doing this frequently is often too computationally expensive. LLaMA [40], for instance, was trained for 21 days on 2,048 A100 GPUs, costing over \$2.4M and emitting over 1,000 tons of $CO_2$. Even repeatedly retraining or finetuning smaller models [3] quickly costs too much for most practitioners. To enable cheap, targeted updates to big, pre-trained models, we study *lifelong model editing*. Here, we continually make targeted edits sequentially throughout a model's deployment. Success means correcting a model's predictions on a stream of edits without decaying its performance on unrelated and previously-edited inputs.

Two possible approaches to lifelong model editing are *continual learning* and conventional *model editing*. While continual learning methods can update models sequentially, when used for sequential editing, they suffer from overfitting [25] and quickly forget previous edits and pre-training data [16, 22]. Alternatively, existing *static* model editors could also be run sequentially. But the editors that

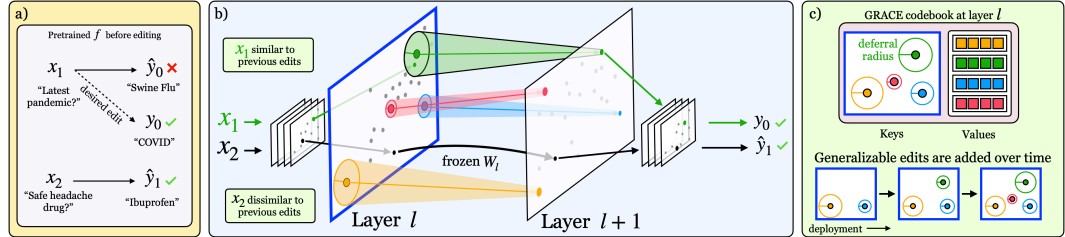

Figure 1: Overview of lifelong model editing with GRACE. a) Models make important errors that must be corrected. b) GRACE makes edits by learning, caching, and selectively retrieving new transformations between layers. c) Edits appear sporadically and require quick fixes, so GRACE codebooks are curated over long sequences of edits.

rely on regularized finetuning [8, 28, 29, 37] succumb to overfitting [4], while hypernetwork methods require pre-training on hard-to-access data [30, 31]. Further, these existing approaches require large sets of representative training edits, access to the model's pre-training data, or semantically-equivalent inputs. Accessing these data is often unrealistic or impractical, and existing editor performance severely decays the pre-trained model after only a few sequential edits [12, 16].

We study lifelong editing using *only* singular inputs to edit models, where models are edited immediately upon arrival. This setup extends beyond recent works, which use many inputs for editing, including collected sets of semantically-equivalent inputs, training edits, and the model's pre-training data. Editing in our realistic, more cost-effective setup is hard because each edit must fix the flagged error, while generalizing to similar future inputs without impacting unrelated model behavior.

To address the challenging lifelong model editing setup, we propose General Retrieval Adaptors for Continual Editing, or GRACE. While we study transformers for natural language processing, GRACE is broadly applicable. As illustrated in Figure 1, GRACE edits a model by adding an Adaptor to a chosen layer, while never changing its weights. This Adaptor then modifies layer-to-layer transformations for select inputs. By caching embeddings for input errors and learning values that decode into desired model outputs, GRACE serves as a codebook in which edits are stored, enabling longer sequences of edits than prior works. To encourage generalization, we leverage the models' semantic similarity in its latent space by introducing $\epsilon$-balls around cached edits. GRACE then only applies for inputs near existing keys, differing from more-traditional Adaptors [15]. By managing the $\epsilon$-balls' size over time, GRACE makes immediate edits, remembers previous edits, and leaves correct model behaviors intact, making it parameter efficient. Further, since GRACE codebooks leave model weights unaltered and are fully model-agnostic, they also point towards plug-and-play, cost-effective model editing, especially to make crucial spot-fixes between bigger retraining efforts.

Our contributions are as follows:

1. We establish key metrics and comparisons for lifelong model editing, an important but understudied and challenging problem setting. We introduce two new public benchmarks for lifelong model editing: mitigating LLM hallucination [27] and addressing label shifts [6].

2. We develop GRACE, a method for lifelong model editing that avoids expensive retraining or finetuning. Unlike other works, GRACE requires no inputs beyond singular edits. GRACE works with autoregressive and non-autoregressive models alike and achieves parameter efficiency without touching the model's weights.

3. Our experiments show that GRACE outperforms seven alternatives when sequentially editing T5, BERT, and GPT models for question answering, document classification, and language generation. We show that GRACE edits can generalize to new inputs without memorizing while incurring only a small, one-time cost to inference time over long sequences of edits.

## 2 Methods: Lifelong Model Editing with GRACE

### 2.1 Problem Formulation

The Lifelong Model Editing task is to edit the same model hundreds to thousands of times *in a row* without forgetting upstream performance or fixes for previous edits. Let $f_0$ denote a model with

frozen parameters that was pre-trained on dataset $\mathcal{D}_{\text{train}}$. For clarity, we drop the subscript where possible. Assume that $f$ consists of $L$ layers, where $f^l(\cdot)$ computes the hidden state at layer $l$. In this work, we assume $f$ is a transformer architecture for natural language, though our principles are general. We then deploy $f$ on a stream of inputs $[x_0, x_1, ...]$, where $x_t \in \mathcal{D}_{\text{edit}}$, which contains samples observed during deployment. We then monitor the model's predictions $\hat{y}_t = f(x_t)$ over the stream. Note that the prediction tasks during training and deployment are the same. Over time, the model makes errors such that $\hat{y}_t \neq y_t$, where $y_t$ is the true label. To continue safely deploying $f$, we aim to *edit* $f$ such that $f(x_t) = y_t$. After each edit, the updated $f$ should 1) be successfully edited such that $f(x_t) = y_t$, 2) retain accuracy on prior edits $x_{<t}$, and 3) retain its behavior on its training data: $f(x_i) = f_0(x_i) \, \forall \, x_i \in \mathcal{D}_{\text{train}}$. In contrast to prior works [8, 28, 30, 31, 37], we assume access to only input edits $x_t$ and corrected labels $y_t$, since $\mathcal{D}_{\text{train}}$ is often proprietary or prohibitively large, and collecting training edits or semantically-equivalent inputs is often expensive in practice.

## 2.2 GRACE: General Retrieval Adaptors for Continual Editing

We propose GRACE, a method for sequentially editing a pre-trained model's behavior *without* altering its weights, as illustrated in Figure 1. GRACE works by wrapping a chosen layer of any pre-trained model architecture with an Adaptor. A GRACE Adaptor at model $f$'s layer $l$ contains two components: (1) a codebook $\mathcal{C}$ and (2) a deferral mechanism to decide whether to use $\mathcal{C}$ for a given input.

**GRACE codebook.** A GRACE Adaptor at layer $l$ maintains a discrete codebook, adding and updating elements over time to edit a model's predictions. The codebook contains three components:

- *Keys* ($\mathbb{K}$): Set of keys, where each key is a cached activation $h^{l-1}$ predicted by layer $l-1$.
- *Values* ($\mathbb{V}$): Set of values that are randomly initialized and are updated using the model's finetuning loss for edits. Each key maps to a single, corresponding value.
- *Deferral radii* ($\mathcal{E}$): Each key has a *deferral radius* $\epsilon$, which serves as a threshold for similarity matching. The deferral mechanism uses this radius as shown in Algorithm 1. GRACE is activated at layer $l$ *only* if the deferral constraint is satisfied. New entries have a default value $\epsilon_{\text{init}}$, which is a hyperparameter.

**Deferral mechanism.** Before editing, GRACE layers are empty. As editing progresses, GRACE adds keys and adapts values and $\epsilon$ entries. Conceptually, inference at layer $l$ with GRACE entails a deferral decision, computing $h^l$ using a similarity search over GRACE's keys:

$$h^l = \begin{cases} \text{GRACE}(h^{l-1}) & \text{if } \min_i(d(h^{l-1}, \mathbb{K}_i)) < \epsilon_{i_*}, \text{ where } i_* = \text{argmin}_i(d(h^{l-1}), \mathbb{K}_i), \\ f^l(h^{l-1}) & \text{otherwise,} \end{cases} \quad (1)$$

where $f^l(h^{l-1})$ denotes the *unedited* model's activation of the $l$-th layer. $h^{l-1}$ can be seen as a *query* to the codebook, and $\text{GRACE}(h^{l-1})$ retrieves the value associated with its closest key. $\epsilon_i^l$ and $\mathbb{K}_i^l$ are the influence radius and key $i$ in layer $l$, respectively, and $d(\cdot)$ is a distance function. We follow related work [41] and use Euclidean distance for $d(\cdot)$ in our experiments—changing this is trivial. Through explicit similarity search, we use the fact that large models encode semantic similarity with respect to their tasks in their latent spaces. This lets GRACE edits to generalize to similar inputs in the future. If a new input is unlike any cached keys, GRACE simply defers to $f$'s pretrained weights. This way, GRACE layers limit interference with $\mathcal{D}_{\text{train}}$ by leaving the model weightsunaltered, which helps when input distributions shift [45].

**Codebook maintenance.** To make an edit, a GRACE layer can perform one of two operations. Each step is described in Algorithm 1. First, if the codebook is empty or the input embedding $h^{l-1}$ falls *outside* the deferral radius of all existing keys according to distance function $d(\cdot)$, then a new codebook entry is created and added: $\{(h^{l-1}, v, \epsilon_{init}, y)\}$. Thus if $x_t$ were passed into $f$ again, $h^{l-1}$ would activate the codebook and value $v$ would be passed to layer $l+1$. Training $v$ is detailed below.

Sometimes, a query $h^{l-1}$ will be close enough to an existing key that adding a new entry would cause their $\epsilon$-balls to overlap. To avoid this, we compare the edit label $y$ to the model's prediction for the nearest key and distinguish two cases: 1) If the overlapping key's label is the *same* as the edit's label, **Expand** that key's $\epsilon$ to encompass the query. 2) If the overlapping key's label is *different* from the edit's label, **Split** these keys by first decreasing the influence radius of the overlapping key, then adding a new codebook entry where the new key is simply the query $h^{l-1}$. We set both keys' $\epsilon$ values to be half their distance apart.

As edits stream over long deployments, by continuously adding and updating the GRACE codebook, layer $l$'s latent space is partitioned according to which inputs needed edits. When *not* performing edits, these codebook maintenance operations are bypassed, and keys are entirely frozen. GRACE thus introduces a new model editing paradigm in which edits can be made sequentially, similar edits are encouraged to be edited similarly, and the ultimate influence of new edits can be controlled and monitored explicitly. $\epsilon_{\text{init}}$ is the sole parameter in GRACE, which sets the initial $\epsilon$ value for new codebook entries. Intuitively, using a larger $\epsilon_{\text{init}}$ will create edits with more influence, making edits more general, but increasing the interference with unrelated inputs. In practice, $\epsilon_{\text{init}}$ could be tuned using either exogenous data or GRACE codebooks can periodically be refreshed.

**Training GRACE Values.** When making an edit with GRACE, either a new key–value pair is learned or an existing key–value pair is updated. To ensure that newly-learned values correct the model's behavior, we train them directly using backpropagation through the finetuning loss on the model's prediction given the edit. The learned value $v$ then replaces $h^l$ for the rest of the forward pass. In our experiments, we train values using 100 gradient descent steps to train the values and ensure the model's behavior is updated.

---

**Algorithm 1:** Update Codebook at layer $l$.

**Input:** $\mathcal{C} = \{(\mathbb{K}_i, \mathbb{V}_i, \epsilon_i)\}_{i=0}^{C-1}$, codebook
**Input:** $f(\cdot)$, model
**Input:** $y_t$, desired label
**Input:** $x_t$, edit input for which $f(x_t) \neq y_t$
**Input:** $\epsilon_{\text{init}}$, initial $\epsilon$
**Input:** $d(\cdot)$, distance function
**Output:** $\mathcal{C}$, updated codebook
$C = \|\mathcal{C}\|$
$\hat{y}, h^{l-1} = f^L(x_t), f^{l-1}(x_t)$
$d_{\min}, i = \min_i(d(h^{l-1}, \mathbb{K}_i))$
If $d_{\min} > \epsilon_i + \epsilon_{\text{init}}$ or $C = 0$:
  # $h^{l-1}$ far from existing entries or empty $\mathcal{C}$
  $v_{\text{new}} = $ finetune on $P_f(y|v_{\text{init}})$
  $\mathcal{C}_C = (h^{l-1}, v_{\text{new}}, \epsilon_{\text{init}})$ # *Add entry*
Else:
  # $h^{l-1}$ near existing entries
  If $f^L(k_i) = y$:
    # *Same label → Expand*
    $\mathcal{C}_i := (k_i, v_i, \epsilon_i + \epsilon_{\text{init}})$
  Else:
    # *Different label → Split*
    $\mathcal{C}_i = (k_i, v_i, d_{\min}/2)$ # *Update entry i*
    $v_{\text{new}} = $ finetune on $P_f(y|v_{\text{init}})$
    $\mathcal{C}_C = (h^{l-1}, v_{\text{new}}, d_{\min}/2)$ # *Add entry*
**return:** $\mathcal{C}$

---

**GRACE layers with sequential inputs.** For models with different representations per input token, like transformers, we must choose 1) which token should be GRACE's input query, and 2) which token to replace with a retrieved value in the subsequent layer. In practice, we find that broadcasting the value to each token is reasonable for tasks like classification, since values gain strong control over the model's behavior and makes them easy to learn. For autoregressive models however, picking the right tokens for the query and value is more nuanced. We opt for replacing only the final token of an input prompt, which we verify experimentally, since compressing future generated text into tokens is an interesting and burgeoning direction in itself [32]. Upon choosing these tokens, GRACE naturally applies to all popular transformer models.

## 2.3 Illustrative Example

To garner intuition about GRACE, we provide an illustrative experiment on synthetic data. As shown in Figure 2, we sample 100 instances from two 2D distributions corresponding to classes. We then train a three-layer binary classifier with two 100-dimensional hidden layers and ReLU activations. Next, we introduce edits with flipped labels, simulating local label shift at test time. Using a single

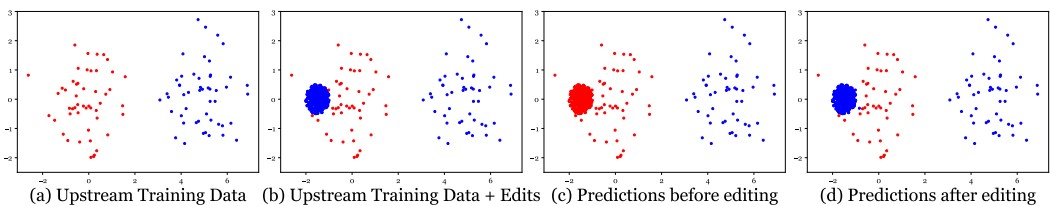

(a) Upstream Training Data    (b) Upstream Training Data + Edits    (c) Predictions before editing    (d) Predictions after editing

Figure 2: Illustrative example of GRACE. We train a model on separable data in (a), then introduce locally-flipped labels at test time in (b). In (c), the original model unsurprisingly misclassifies these label-flipped instances. In (d), GRACE fixes these labels without impacting other inputs.

| SETTING | MODEL | TEST RETENTION DATA | | | EDITING DATA | | |
|---------|-------|---------|---|---------|---------|---|---------|
| | | Dataset | N | Pre-edit | Dataset | N | Pre-edit |
| QA | T5 (60m) | NQ [20] | 1000 | .72 F1 | zsRE [24] | 1000 | .31 F1 |
| Clf. | BERT (120m) | SCOTUS$_{1982-1991}$ [6] | 914 | .99 Acc | SCOTUS$_{1992-2009}$ [6] | 931 | .55 Acc |
| Halluc. | GPT-2 (1.5B) | OpenWebText [10] | 1000 | 15.98 PPL | SelfCheckGPT | 1392 | 132.7 PPL |

Table 1: Dataset statistics for main results. *Test retention* is the testing set of each model's training data. *N* is the number of samples. *Pre-edit* is the unedited model's performance on each dataset.

key in layer two, GRACE can successfully correct these errors while barely influencing other inputs. Meanwhile, finetuning on these errors alone will clearly break the model.

## 3 Experiments

### 3.1 Experimental Setup

We evaluate GRACE's capacity to edit models hundreds to thousands of times sequentially. This is in contrast to prior works, which largely evaluate using single edits. Further, much of the model editing literature relies on synthetic edits, often generated by flipping random labels. Beyond unrealistically assessing performance, it is well-established that learning random versus natural labels is vastly different [49]. Instead, we correct authentic mistakes made by models, each time an error is made.

**Baselines.** We compare against continual learning and model editing methods. First, we continually finetune (**FT**) [25] on streaming errors. To reduce overfitting, we also compare with Elastic Weight Consolidation (**EWC**) [19] and a simple form of experience replay [36] where we periodically retrain the model (**Retrain**) on all previous edits. Second, we compare against model editors **MEND** [30] and **Defer**, inspired by SERAC [31] but altered for our setup. We also compare against **ROME** [28] in our language modeling experiment. Finally, we ablate GRACE by replacing our discrete search with a **Memory** network containing memory module that is indexed by a soft attention mechanism. Details for all comparisons are in Appendix C.

**Datasets and Pre-trained Models.** We evaluate GRACE on three sequential editing tasks with corresponding pre-trained models, as shown in Table 1. 1) We edit a 60-million parameter T5 model [35] trained for context-free question-answering, as is used in [30]. We extract potential edits from the validation set of **zsRE** [24], following the editing literature [16, 30]. This model achieves F1 of .72 on NQ and .31 on zsRE prior to editing. Further details are in Appendix B.1. 2) We edit a 110-million BERT classifier trained for a new editing task with label shift using the **SCOTUS** dataset from Fairlex [6]. The prediction task is to categorize U.S. Supreme Court documents over multiple decades into 11 topics. Over time, categorization rules change, so label distributions shift. We train a BERT classifier on the 7.4k training cases from 1946-1982, then make edits on 931 cases from 1991-2009. We also introduce additional, realistic label shifts, as discussed in Appendix B.2. This model achieves Accuracy of 0.99 on the training set and .55 on the edit set. 3) We introduce a new editing task by correcting a GPT language models' **Hallucination**. In [27], authors prompt GPT-3 to generate 238 wikipedia-style biographies using subjects from WikiBio. They then annotate the factual accuracy of each sentence, recording which are hallucinations. We propose editing inaccurate sentences by replacing them with corresponding sentences in the true wikipedia entries. We include edits for all 238 biographies, creating 1392 sequential edits and 592 already-accurate outputs. We then edit GPT2-XL, which has 1.5B parameters. However, GPT2-XL was not trained on this task, so has high perplexity (PPL) on all sentences. Therefore, we finetune GPT2-XL on these data mixed with sentences from OpenWebText [10], a public version of GPT2's training data. Our final GPT2-XL model has PPL of 15.98 on OpenWebText (comparable to the original), 8.7 on already-accurate outputs, and 132.7 on intended edits, indicating a need for editing and room for improvement. Further details are available in Appendix B.3.

**Metrics.** To compare each method, we measure three main metrics that align with prior work [16, 28].

1. Edit Success (**ES**): We check whether an edit has been successful using ES: $m(y, \hat{y})$, where $m(\cdot)$ is a task-specific measure of accuracy: standard F1 for question answering, Accuracy for classification and Perplexity (PPL) for generation.

| Method | zsRE (T5; F1 ↑) | | | | SCOTUS (BERT; Acc ↑) | | | | Hallucination (GPT2-XL; PPL ↓) | | | | |
|---|---|---|---|---|---|---|---|---|---|---|---|---|---|
| | TRR | ERR | *Avg.* | *#E* | TRR | ERR | *Avg.* | *#E* | TRR | ERR | ARR | *#E* | time (s) |
| FT [25] | .56 | .82 | *.69* | 1000 | .52 | .52 | *.52* | 415 | 1449.3 | 28.14 | 107.76 | 1392 | .26 (.07) |
| FT+EWC [19] | .51 | .82 | *.66* | 1000 | .67 | .50 | *.58* | 408 | 1485.7 | 29.24 | 109.59 | 1392 | .29 (.06) |
| FT+Retrain [36] | .27 | .99 | *.63* | 1000 | .67 | **.83** | *.75* | 403 | 2394.3 | 35.34 | 195.82 | 1392 | 23.4 (13.2) |
| MEND [30] | .25 | .27 | *.26* | 1000 | .19 | .27 | *.23* | 672 | 1369.8 | 1754.9 | 2902.5 | 1392 | .63 (.10) |
| Defer [31] | **.72** | .31 | *.52* | 1000 | .33 | .41 | *.37* | 506 | 8183.7 | 133.3 | 10.04 | 1392 | .07 (.02) |
| ROME [28] | — | — | — | — | — | — | — | — | 30.28 | 103.82 | 14.02 | 1392 | .64 (.28) |
| Memory | .25 | .27 | *.26* | 1000 | .21 | .20 | *.21* | 780 | 25.47 | 79.30 | 10.07 | 1392 | .11 (.02) |
| GRACE | .69 | **.96** | ***.82*** | 1000 | **.81** | .82 | ***.82*** | 381 | **15.84** | **7.14** | **10.00** | 1392 | .13 (.02) |
| | *137 keys (7.30 edits/key)* | | | | *252 keys (1.51 edits/key)* | | | | *1341 keys (1.04 edits/key)* | | | | |

Table 2: Comparison of GRACE to existing methods. Metrics shown are computed after all sequential edits. For Hallucination, we also compute perplexity retention on already-accurate sentences (ARR) and the average time per edit. Parentheses in *time* denote standard deviation. We also count the number of keys GRACE used. GRACE achieves the best TRR/ERR balance while making edits faster than most comparisons and using small codebooks for zsRE and SCOTUS. A large codebook is required for Hallucination, since each edit label is unique.

2. Test Retention Rate (**TRR**): We check how well an edited model retains its performance on its original testing data using TRR: $\frac{1}{n}\sum_{i=1}^{N} m(f(x_i), y_i)$, where $(x_i, y_i) \in \mathcal{D}_{\text{test}}$, $f$'s original test set. For T5, $\mathcal{D}_{\text{test}}$ is 1k random samples from NQ, for our BERT model, $\mathcal{D}_{\text{test}}$ is court documents from 1982-1991, and for GPT2-XL, $\mathcal{D}_{\text{test}}$ is WebText, for which we use the first 1k sentences of OpenWebText [10].

3. Edit Retention Rate (**ERR**): We check how well an edited model retains previous edits through ERR: $\frac{1}{n}\sum_{i=1}^{N} m(f(x_i), y_i)$ where $(x_i, y_i) \in \mathcal{D}_{\text{edits}}$.

We also track the number of edits (**#E**), which may vary by editor as different editors will lead to different mistakes in practice. Further implementation details are available in Appendix A.

## 3.2 Results

### 3.2.1 Comparisons to existing methods

We first find that GRACE outperforms existing methods after long sequences of edits, as shown in Table 2. We focus here on TRR and ERR, which have a trade-off as each can be achieved in isolation, though all metrics are reported in Appendix G. On zsRE and SCOTUS, we focus on TRR, ERR, and their average. ROME is incomparable on these dataset as it is only proposed for GPT models. On Hallucination, we include Accurate Retention Rate (ARR), which is the edited model's perplexity on sentences on which it was already accurate.

On zsRE and SCOTUS, GRACE's averaged TRR and ERR outperforms its closest competitors by 19% and 9%, respectively. Comparing the average performance across all methods, GRACE's improvement is 86% and 129% due to other methods' inability to balance ERR and TRR. For instance, while Defer achieves the highest TRR on SCOTUS it trades off ERR. In contrast, FT+Retrain achieves the highest ERR on SCOTUS, but trades off TRR. On both datasets, MEND and Defer both struggle to balance TRR and ERR without access to privileged data.

GRACE also outperforms the comparisons on the Hallucination task. We observe that the finetuning methods easily overfit to new edits, leading to competitive ERR but poor TRR. However, their good ERR comes at the expense of ARR in these methods. We also find that ROME and Memory both perform competitively with GRACE, indicating the value of parameter efficiency in lifelong model editing. Finally, we report the average wall-clock time it takes to perform one edit, finding that GRACE makes edits twice as fast as finetuning while competing with other Adaptors.

Excitingly, GRACE can achieve high-quality edits with small codebooks. For example, GRACE uses few keys to edit T5 on zsRE: 1000 edits are made using only 137 keys. Such compression is only feasible by managing the $\epsilon$-balls effectively over time. This is also parameter-efficient: with details in Section 3.2.3, the zsRE codebook contains 210,569 scalar values (.35% of T5's parameters) and

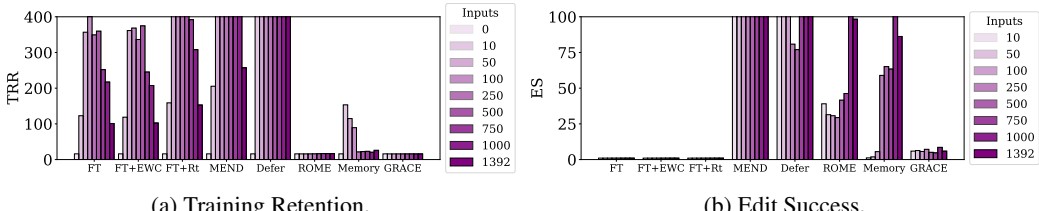

(a) Training Retention.  (b) Edit Success.

Figure 3: ES and TRR while editing GPT2-XL on Hallucination. Lower values are better because TRR and ERR measure perplexity. GRACE outperforms the comparisons by making successful edits while maintaining the model's training knowledge. All metrics are shown in Figure 13.

only 70,281 are learnable. The number of keys is also lower-bounded by the number of unique edit labels and is related to the complexity of the inputs. On SCOTUS, GRACE also achieves a relatively small codebook, showing a balance between input complexity and generalizability of each edit. Since SCOTUS is a document classification task with 11 classes, there could be as few as 11 keys, but given each document contains many sentences, there is a lower chance that the same keys can be used for many documents. That GRACE's keys represent 1.51 edits on average indicates significant generalization. As expected, the lower-bound on codebook size is evident in the Hallucination task, where GRACE uses a new key for almost every edit. This is because edit labels are unique sentences. Despite a large codebook, such success at selectively inserting tokens that generate full sentences poses an exciting direction for controllable and editable language modeling.

In Figure 3, we show a more-detailed comparison of all methods for the Hallucination editing task over time, focusing on TRR and ES (see Figure 13 for all metrics). As expected, finetuning methods excel at ES, but perform poorly at TRR. On the flip side, ROME and Memory have reasonably-low ES and TRR scores, though both suffer as edits progress. While ROME is competitive early in editing, it especially suffers on ERR (Table 2) and GRACE's TRR remains 2x better after making all edits.

### 3.2.2 Model Analysis: Memorization vs. Generalization

Next, we study GRACE's memorization vs. generalization performance by editing T5 on zsRE on extremely long sequences of edits while varying $\epsilon_{init}$ and the edited layer. We split each zsRE question's set of rephrasings into two sets: edits and holdouts. Then, we pass edits through GRACE until 3,000 edits are made. This is extremely large: even 10 edits catastrophically decays performance [12] and [16] performs roughly half as many edits. After each edit, we measure TRR, ERR, F1 on the entire Holdout set, and record the number of keys. Since we evaluate GRACE on the entire Holdout set after each edit, the results start low since has seen no rephrasings of held out edits. Therefore, later measures of Holdout performance are more representative than earlier. Figure 4 shows our results for $\epsilon_{init} = 0.1$ and $\epsilon_{init} = 3.0$, while the rest are in Appendix H. We derive the following findings.

**Layer and $\epsilon_{init}$ choice balance memorization and generalization.** We first find that different blocks lead to different editing performance. Notably, editing Blocks two and four achieve high TRR, high ERR, and strong generalization for both choices of $\epsilon_{init}$ On the contrary, for Block 6 we see that since each $\epsilon_{init}$ is small, they do not generalize at all and also lead to poor ERR. As expected, when $\epsilon_{init}$ is small, TRR performance is optimal, since most edits require a new key. While creating a large codebook can be feasible, the trade-off in Holdout becomes clear, as detailed in the next finding. The relationship between layer choice, performance, and the size of the codebook likely stems from a layer's representational capacity: Layers that map semantically-equivalent inputs near one another will be easier to edit with GRACE.

**GRACE edits generalize to unseen inputs.** Steadily increasing Holdout shows that that GRACE edits can generalize to previously-unseen holdout edits. Excitingly, interior layers appear to generalize better than early and late layers, which is backed up by their use of fewer keys. Larger $\epsilon_{init}$ values also lead to better generalization, as expected, implying that the semantically-similar inputs indeed land in the same deferral radii. The later layer, Block 6, appears to

**GRACE codebooks stay small and stabilize over time.** Finally, the number of keys over time steadily flattens, indicating that the $\epsilon$ values indeed adapt to the data distribution. As we increase $\epsilon_{init}$,

the resultant codebooks get smaller, indicating control over codebook size. Small codebooks are also important for parameter efficiency and in generalization.

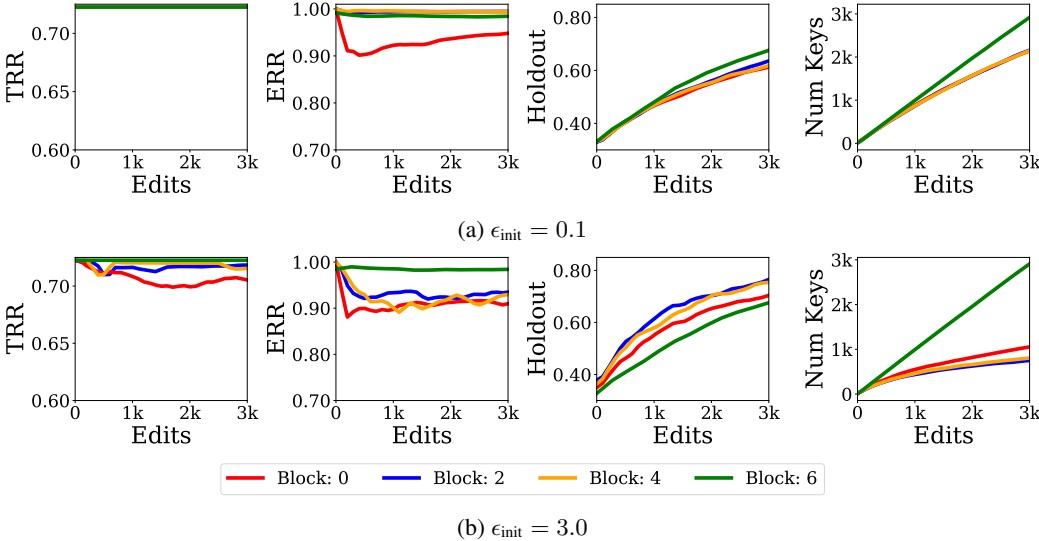

(a) $\epsilon_{\text{init}} = 0.1$

(b) $\epsilon_{\text{init}} = 3.0$

Figure 4: Impact of $\epsilon_{\text{init}}$ and block choice for GRACE editing T5 on zsRE for 3000 sequential edits. Other $\epsilon_{\text{init}}$ values are in Appendix H. Along with TRR and ERR, we also measure F1 on a "Holdout" edit set containing unseen rephrasings of all edits. We find that blocks 0 and 6 use more keys and achieve higher TRR, but can lead to lower ERR and generalize worse, given lower holdout values.

### 3.2.3 Parameter Efficiency

GRACE's memory requirements are small and straightforward: A new edit requires $|h^{l-1}| + |h^l| + 1$ parameters, where $|h^{l-1}|$ is the dimension of the key, $|h^l|$ is the dimension of the value, and $\epsilon_{\text{init}}$ is a scalar. Further, the key's $|h^{l-1}|$ parameters are not trained. As detailed in Appendix E, this is comparable to the alternatives when performing thousands of edits, so performance gain is not from parameter count. This intuition is reinforced by the fact that at inference time, if the Adaptor is activated, predictions are only altered using the $|h^{l-1}| + |h^l| + 1$ parameters of the chosen entry.

### 3.2.4 Interpreting GRACE Codebooks

A key benefit of GRACE is that learned codebooks can be detached from the model and inspected. This way, edits can easily be undone without impacting the model and the codebooks can be inspected throughout editing. To demonstrate this advantage, we inspect how keys and their $\epsilon$ change throughout editing Block 4 of the T5 QA model. In this experiment, we investigate how well GRACE edits generalize to unseen edits. At each edit in a sequence of 1,000 inputs, we pass the entire holdout set of edits through the newly-edited model. For each holdout instance, we record in which key's $\epsilon$-ball it lands, if any. This way, we track whether a newly-added key generalizes to multiple holdouts successfully. We also track what proportion of the holdouts land inside *any* key to evaluate generalization over time. In Figure 5, we summarize the results from one such experiment, with more details in Appendix D. We find that the number of holdouts per key stabilizes over time. Interestingly, $\epsilon_{\text{init}}$ values capture too many holdouts per question, which explains the trade-off between generalization and TRR. In the Appendix, we also show that the number of holdouts per key are stable, while others are highly-variable. This implies that our codebook maintenance strategy can have big ripple effects according to the key. Further, some regions of the latent space are likely better than others for performing GRACE edits.

### 3.2.5 Inference Time

To better understand GRACE's limitations, we compare the inference time of the T5-small QA model before and after editing. We compute the time it takes to run one instance through the model for each of 5,000 edits. We edit T5's block 4 and use a small $\epsilon_{\text{init}}$ of 0.1 to encourage a quickly-growing

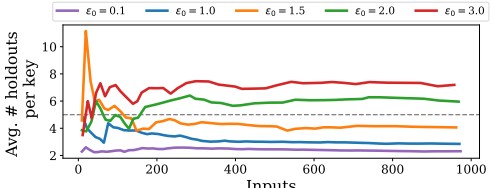

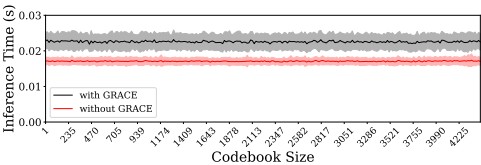

Figure 5: Interpreting GRACE keys throughout editing. Larger $\epsilon_{\text{init}}$ achieve good generalization. The grey line is the true holdouts per edit.

Figure 6: Comparing inference time of GRACE-edited and unedited T5 QA model as the size of the codebook increases.

codebook. To evaluate the variance in training time, we replicate this experiment ten times, each leading to a codebook of roughly 4500 elements. We then average over 20-timestep windows and show standard deviation across all replications.

We find that inference with a GRACE-edited model was only 1.32x slower than an unedited model on average, as shown in Figure 6. This is a one-time cost that remains fixed even as the codebook grows. Inference time is unchanged because search between one query and a full set of keys can be vectorized. Until the codebook outgrows available memory, this cost remains fixed. That GRACE causes any slowdown is a limitation and an avenue for future work.

# 4   Related Work

**Model editing.** Model editing is a new and active research area where the goal is to make targeted changes to a pre-trained model's behavior. Most methods propose variants of regularized-finetuning via auxiliary data, like training instances from the pre-trained model's training data or by using semantically-equivalent versions of new edits [37]. Access to such data is non-trivial, especially as training data and paradigms are becoming proprietary, and collecting semantically-equivalent inputs is often unlikely. Recent works have retained these requirements, while extending to pretrain hypernetworks that predict edits [8, 30, 31], often decomposing weight updates into low-rank components [28, 29]. Recent approaches all study transformer architectures due to their popularity and high training cost [48, 51]. To ensure targeted edits, recent methods like MEND [30] and ROME [28] also draw inspiration from parameter-efficient finetuning [15]. But such methods are known to often require more finetuning steps and are prone to overfit more than regular finetuning [39, 50]. Further, recent works show that edited models are deeply fragile [4] and methods for picking which parameters to update are surprisingly unreliable [13].

Unfortunately, nearly all model editing works consider only *static* edits, making only one edit to a model. While some recent works like MEMIT [29] indeed perform multiple edits, they do so simultaneously, not over time during deployment. Other works demonstrate that editing performance decays quickly when making multiple edits [30], though recent works like SERAC [31] and MEMIT [29] show burgeoning results in this direction. However, these exciting works still use large amounts of privileged information. Most similar to our work, two recent papers discuss sequential editing. First, [12] shows that after editing the same model 10 times, editing performance drops dramatically. Second, a concurrent paper [16] recently proposed a sequential editing setup. However, their method and implementation is architecture-specific and relies on large sources of unrelated inputs.

**Continual Learning.** Continual learning methods are a reasonable approach to lifelong model editing. Most-similar to our setup, recent works have investigated continual finetuning, where large language models are refined over time as new instances arrive. For example, [25] perform a large benchmark of continual finetuning. They find that regularizing finetuning with continual learning methods like Elastic Weight Consolidation [19], Experience Replay [36], and Maximally Interfered Replay [1], quickly decays performance on prior tasks, though it helps to remember some previous inputs. This implies that *editing*, as opposed to regular continual finetuning, is particularly challenging, since edits are unlikely to be uniformly distributed [14]. One promising avenue for continual learning is key-value methods, stemming from computer vision [26, 34, 42] For example, recent works have demonstrated continual prompt-learning for NLP [44, 45] for applications like text retrieval [47]. Recent works have shown that *discrete* key-value methods in particular perform well with shifting distributions [41], with recent works extending to question answering [7]. By caching values, these approaches keep inputs in-distribution for downstream encoders while opening doors to longer-term

memory, resources permitting. We corroborate these advantages in our experiments, where we demonstrate GRACE's robustness to shifting inputs and labels over long sequences of edits.

## 5   Limitations and Ethical Considerations

Lifelong model editing is new and challenging, so our method GRACE has limitations. Understandably, adding similarity search to the layers of a model will slow down inference. Still, while we do not emphasize inference time, accelerating GRACE is natural future step that has been successful in similar methods [15, 33]. Future works may also scale GRACE up to multi-layer edits, a setting unconsidered in this work. While GRACE has already scaled up continual editing to ∼5k edits, real world scenarios might entail approaches that work at an ever larger editing scale, which presents a promising direction of future work. Another limitation of GRACE-style edits is *implication*: Behaviors are edited in isolation. For example, if we edit a model's knowledge of the *latest pandemic in the U.S.* from Swine Flu to COVID, its knowledge of *the second-to-last pandemic* will not be updated.

While editing models can improve their behavior, it can surely be used for harm. For example, a bad actor might edit a LLM to *increase* hate. This limitation is true for all model editors, GRACE included. However, most model editors today directly update the model's weights. This makes it hard to trace back what edits have been made to a model and understand their impact. Transparent editors like GRACE are a promising direction for overcoming this problem. For example, GRACE's codebook can be directly inspected to see what predictions stem from each value and examine properties of the latent space covered by the $\epsilon$-balls.

## 6   Conclusions

Pre-trained models continue to grow and are being applied to a diverse set of downstream tasks. However, they still misbehave in unpredictable ways when deployed. Correcting such behavior is a challenging problem, especially when only some of the model's behavior is problematic. While regularized finetuning or retraining on better data can mitigate this problem somewhat, big models remain too expensive to train for most researchers. We instead build on the model editing literature, and study *Lifelong Model Editing*, an important but understudied problem. With a focus on transformer models for natural language processing, we edit models thousands of times in a row as edits stream during simulated deployments. Our edits only use singular, authentic errors, in contrast to recent works which make synthetic edits and require large amounts of exogeneous data like sets of training edits, semantically-equivalent examples, or pre-training data. We then present GRACE, a plug-in Adaptor for a model's chosen layer that leaves the trained weights untouched. GRACE Adaptors (1) retain the functionality of the original model, while (2) successfully editing model predictions without forgetting previous inputs. Using three real-world datasets, we edit T5, BERT, and GPT models thousands of times and find that GRACE significantly outperforms seven state-of-the-art alternatives. We further investigate GRACE's capacity to make extremely long sequences of edits and show that GRACE can generalize its edits to unseen inputs, avoiding sheer memorization.

## 7   Acknowledgements

We are grateful to all the support received while conducting this research. This project was supported by Quanta, and Marzyeh Ghassemi was supported by the Herman L. F. von Helmholtz Career Development Professorship and the CIFAR Azrieli Global Scholar award. Yoon Kim was supported by MachineLearningApplications@CSAIL.

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

# A    Implementation Details

**Training Details.** All methods are optimized using Adam [18]. Since edits are singular and sequential in our setup, the batch size is always 1. We trained all methods using various GPUs including 48GB NVIDIA RTX A6000s, 40GB NVIDIA A100s, and 80GB NVIDIA A100s. Timing experiments are reported from experiments using a 48GB NVIDIA RTX A6000 GPU. GRACE does not depend on the scale of the model, but the scale of the model depends on available compute resources. To avoid sharding, we use models that fit on one GPU, thought GRACE principles apply beyond this setup. For Adaptor-based editors, we use 100 iterations of gradient descent per input. For experiments using SCOTUS for classification and zsRE for Question Answering, we include early stopping once the model's output is corrected.

**Hyperparameters.** For our Finetuning, Finetuning with EWC, Finetuning with perioding retraining, Memory, Defer, and MEND comparisons, we consider learning rates of $1.0$, $1e^{-1}$, $1e^{-2}$, $1e^{-3}$, $1e^{-4}$, and $1e^{-5}$. We find that Finetuning, Memory, and MEND worked best with $1e^{-2}$. For Adaptor methods GRACE and Defer, we found that a large learning rate of $1.0$ was required to update the model's behavior since their only control on the model's output is by replacing tokens at one layer.

Choosing which layer to edit is another hyperparameter for all editors. In all of our comparisons between editors, each editor edits the same layer. For T5, this is last dense layer of the last encoder block (`encoder.block[7].layer[1].DenseReluDense.wo`), for BERT it is the second-to-last layer (`bert.encoder.layer[10].output.dense`), and for GPT2-XL we edit the fully-connected component of the thirty-sixth layer (`transformer.h[35].mlp.c_fc`). Layer 35 is not random: layers later than 35 led to less-consistent edit success. This is supported by recent work showing the impact of choosing the right layers to finetune [23]. We study these layer choices below, but note that choosing this layer is a hyperparameter in practice: for the purposes of comparison, we ensure editors are compared when editing the same layers.

GRACE has one unique hyperparameter, $\epsilon_{\text{init}}$, which is the size of the $\epsilon$-ball initialized around new codebook entries. In our main results, we set $\epsilon_{\text{init}} = 0.5$ for zsRE (leads to 137 keys/1000 edits), $\epsilon_{\text{init}} = 1.0$ for SCOTUS (leads to 252 keys/375 edits), and $\epsilon_{\text{init}} = 1.0$ for Hallucination (leads to 1341 keys/1392 edits). We study the impacts of choosing $\epsilon_{\text{init}}$ further in Appendices F and H.

**Runtime.** Our main results report runtime for each method on the Hallucination dataset editing GPT2-XL. This time recorded is *only* the average time required to make one edit. This ignores all preprocessing, precomputation, or pretraining.

# B    Additional Dataset and Model Descriptions

As summarized in Table 1, we provide detailed summaries of the datasets and models we use for editing in our experiments.

## B.1    Editing T5 on zsRE Question Answering

**zsRE Dataset.** zsRE [24] is a context-free question-answering dataset that has been extensively studied in the model editing literature [8, 16, 30, 31]. The same as previous works, we extract our set set from the same validation dataset. For our main experiments, our edit set is the first 200 question–answer pairs in the zsRE validation set that have at least 5 rephrasings each. We then use the first 5 rephrasings for each, leading to 1000 possible edits. To ablate design choices in GRACE, we use these same edits in Appendix F.

We also analyze GRACE's generalization (Table 2 in the main paper) and performance on extremely long sequences of edits (Figure 4 in the main paper and Appendix H) using zsRE. To do this, we repeat the above procedure but use the *10* rephrasings from *1000* question–answer pairs. We then split this into an edit set and a holdout set randomly, leading to 5000 potential edits and 5000 holdout edits.

**T5 QA Model.** We edit the publicly-available 60-million parameter version of T5 (`google/t5-small-ssm-nq` on Huggingface). This model was originally trained on the Natural Questions dataset, of which we use 1000 random samples to evaluate TRR during editing. This model achieves F1 of 0.72 on NQ and .31 on zsRE prior to editing. In our

main results in Table 2, we edit T5's "wo" module in its encoder's 4th block's layer 1: `encoder.block[4].layer[1].DenseReluDense.wo`.

## B.2 Editing BERT on SCOTUS Classification

**SCOTUS Dataset.** We consider a single-label multi-class classification task from Fairlex [6]. SCO-TUS contains supreme court rulings on controversial issues [38]. Given a document describing the court's opinion, the task is to predict to which of 14 topics (classes) the document belongs. The 14 topics classes are clustered according to the matter of dispute: `Criminal Procedure`, `Civil Rights`, `First Amendment`, `Due Process`, `Privacy`, `Attorneys`, `Unions`, `Economic Activity`, `Judicial Power`, `Federalism`, `Interstate Relations`, `Federal Taxation`, `Miscellaneous`, and `Private Action`. There are 7.4k training documents from cases that took place from 1946-1982. Then, there are 914 validation documents from cases from 1982–1991. Finally there are 931 testing cases from 1991–2009. Since annotation guidelines do shift over time [38], we exacerbate this shift slightly by making three realistic changes *only to the training set*:
- Relabel `First Amendment` as `Civil Rights`.
- Relabel `Due Process` as `Civil Rights`.
- Relabel `Unions` as `Economic Activity`.

This realistic type of relabeling creates cases in these data that are especially challenging to forecast during training or adjust for during editing. We perform edits on the test set from 1991–2009, for which no relabeling was done. TRR is computed on the 914 validation documents from 1982–1991.

**BERT.** We finetune a 110 million parameter BERT model (`bert-base-cased` on Huggingface) on the SCOTUS training using Huggingface. We use 50 epochs with a batch size of 4 and a learning rate of $5e^{-05}$ with an AdamW optimizer using Huggingface's default model trainer. The finetuned BERT model achieves 0.99 Accuracy on its original validation data and 0.55 on the edit set, indicating a large amount of improvement to be made from editing. In our main results in Table 2, we edit BERT's "dense" module in its 10th layer: `bert.encoder.layer[10].output.dense.weight`.

## B.3 Editing GPT2 on Hallucination Language Modeling

**Hallucination with SelfCheckGPT.** We examine how well model editors can mitigate hallucination in autoregressive language models using a new dataset released with SelfCheckGPT [27]. To generate this dataset, which we refer to as "Hallucination," the authors prompt GPT-3 to generate wikipedia-style biographies for concepts extracted from WikiBio. They then annotate factual accuracy of each sentence, noting which are hallucinations. We propose editing highly-inaccurate sentences by replacing them with corresponding sentences in the true wikipedia entries. To acquire "true" wikipedia entries, we extract wikipedia summaries from WikiBio and acquire the correct sentence at the same index as the edited sentence. For example, if sentence 3 of a biography of "Bon Jovi" is a hallucination, we would edit a model to produce sentence 3 of the wikipedia entry for "Bon Jovi" given sentences 1 and 2 generated by GPT-3. This way, every edit contains a prompt, which is the GPT-3-generated text up until the hallucination, and a label, which is the correct next sentence from wikipedia. There are 1392 potential edits and 516 already-accurate sentences. The editing task here is to decrease the LLM's perplexity (PPL) on the corrected sentences without increasing its perplexity on either the model's training data or the already-correct portions of Hallucination.

This task is different from prior editing works in two ways: First, these are authentic mistakes made by a high-quality LLM. Other works instead create synthetic edits by asking different language models to generate different inputs and swap between them. As their datasets are generated using far lower quality models, the sentences are highly unrealistic. Second, our edits are substantially longer than prior works, which have 10-token prompts and 10-token sentences to generate. By extending this paradigm to longer sentences, our editing task poses a more realistic view of model editing success.

**GPT2-XL.** Since GPT-3 generated the Hallucination dataset, it would be ideal to edit GPT-3 directly. However, since it is proprietary and too large for practical research, we edit GPT2, using the 1.5B-parameter "XL" model (`gpt2-xl` on Huggingface). Being a "small" LLM, this model is accessible to a broad range of researchers yet is high-enough quality to serve as an effective stand-in for bigger models. Unfortunately, GPT2 is not already a high-quality biography-generator, and so it finds *all* of the Hallucination sentences to be unlikely, even those that are "already correct" from GPT-3. To overcome this, we finetune GPT2 on the already-accurate Hallucination data mixed with 200 random sentences from OpenWebText [10], which is a public version of GPT2's original training

data. This finetuned model achieves perplexity of 15.98 on OpenWebText (comparable to GPT2-XL on its training data), 8.7 on already-accurate outputs, and 132.7 on intended edits. Our finetuned GPT2-XL model will be made public with our paper. Thus with GPT2-XL mimicking GPT-3 for these data, we can evaluate model editors on decreasing the edit PPL while maintaining its PPL on OpenWebText and the already-accurate sentences. In our main results in Table 2, we edit GPT2-XL's "c_fc" module in its 36$^{\text{th}}$ layer: `transformer.h[35].mlp.c_fc`.

## C   Descriptions of Compared Editors

**Finetune.** We finetune the weights in a chosen layer using Adam optimization, leaving all other layers of the pretrained model frozen. Finetuning is done using the Cross Entropy loss for T5 and BERT, and by minimizing the log probability of corrected sentences for GPT.

**Finetune with EWC.** We perform the same finetuning as above, except update the weights with Elastic Weight Consolidation. This is a well-studied approach to minimize catastrophic forgetting in continual learning by regularizing weight updates to leave important weights unchanged over time.

**Finetune with Retraining.** We perform the same finetuning as above, except periodically finetuning the original pre-trained model on all previous edits. This requires caching previous edits and may often exacerbate catastrophic forgetting on the model's pre-training data.

**MEND [30].** MEND uses a hypernetwork to forecast new weights for a pre-trained model's chosen layer by predicting low-rank decomposition of the layer's weight matrix. The hypernetwork is trained on a set of training edits, which include a new edit, a set inputs that are semantically equivalent to the edit, and samples from the model's pre-training data (or some analogous set). The network is trained to make the edit, apply the fix to the semantically-equivalent inputs, and minimize the KL divergence between the new and old model predictions on the model's pre-training data. In our setup, we only have single edits that stream in, so there are neither training edits for the hypernetwork nor access to the model's pre-training data. Therefore, we train the hypernetwork to predict updated weights as edits stream in using continuous finetuning. Per the original paper's recommendations, we train the hypernetwork to predict a new gradient vector and value vector, which are multiplied to create a matrix with which to update the model's existing weights. We use the authors' original code from github.com/eric-mitchell/mend.

**ROME [28].** ROME identifies weights in chosen layers of GPT models and updates them to alter a GPT model's factual knowledge. Being designed only for GPT models, we evaluate this approach on the Hallucination editing task. We use the authors' original code github.com/kmeng01/rome and use a template for wikipedia bios: `This is a Wikipedia passage about {}`.

**Defer.** Inspired by SERAC [31], we implement a deferral-based model editor, which we call Defer. At a chosen layer, for a new input Defer chooses to either 1) trust the pre-trained model's predictions, or 2) replace the layer's prediction with a vector predicted by a new model. This involves learning two new predictors for a Defer Adaptor at a given layer. First, a probability of deferral is predicted by one network $g : \mathbb{R}^{|h^{l-1}|} \rightarrow [0, 1]^1$. For $g$, we use a one-layer network with a sigmoid activation function. Second, a new value is predicted by another one-layer network $o : \mathbb{R}^{|h^{l-1}|} \rightarrow \mathbb{R}^{|h^l|}$. In early experiments, we found that increasing these networks' depth overfits quickly. $g$ and $o$ are jointly trained via the pre-trained model's finetuning loss continually as new edits arrive.

**Memory.** As a softer version of GRACE's codebook, we implement an Adaptor editor based on memory networks [46]. This editor contains two components: A memory module that serves as a predicted cache for learned activations and an attention mechanism that learns to retrieve appropriate memories. We implement the memory module as a matrix $M \in \mathbb{R}^{N \times h^l}$, containing $N$ memories, each of which has $h^l$ dimensions. For our experiments, we set $N = 50$ and the memory module is learnable over time. Then, we learn a simple attention mechanism $g : \mathbb{R}^{h^{l-1}} \rightarrow [0, 1]^{1 \times N}$ where $g$'s outputs sum to one. For $g$, we use a one-layer neural network with a softmax activation function. Then, the editor layer's output is their multiplication $g(h^{l-1})M$, resulting in a newly predicted $|h^l|$-dimensional vector to be passed into layer $l + 1$. $M$ and $g$ are trained jointly via the pre-trained model's finetuning loss continually as new edits arrive.

| Parameter | Search Space |
|---|---|
| Edited Block | 2, 7 |
| $\epsilon_{\text{init}}$ | 0.1, 0.5, 1.0, 3.0 |
| Replacement Token | Last Input Token, All Input Tokens, All Tokens |
| Value Initialization | Uniform: $\mathcal{U}(0,1)$ ("cold"), $h^l$ ("warm") |

Table 3: Hyperparameter options we searched through for GRACE on T5 for zsRE edits.

## D   Interpreting GRACE codebooks

As introduced in the main paper in Section 3.2.4, a benefit of GRACE is that the codebook is detachable from the pretrained model. This feature allows for closer inspection of the editing process. We first expand the results in the main paper by including the average F1 score per key in Figure 7, where we see that keys overall achieve highly-accurate edits on holdout data. This implies that when a holdout question has a key, the key maps to the correct answer. However, not all holdouts have associated keys. We therefore examine this further in Figure 8, where we investigate the top-10 keys containing the largest numbers of holdout questions for 5 $\epsilon_{\text{init}}$ choices. As expected, bigger $\epsilon_{\text{init}}$ lead to more holdouts per key. Meanwhile, small $\epsilon_{\text{init}}$ values lead to very few holdouts being captured by each key. Additionally, the big $\epsilon_{\text{init}}$ values lead to small codebooks and small values lead to big codebooks. So when $\epsilon_{\text{init}} = 0.1$, 60% of the holdouts do eventually have a key, but this is at the expense of codebook size, which grows to 547 entries. On the right-hand side, we also see the proportion of the holdout set that has *any* associated key.

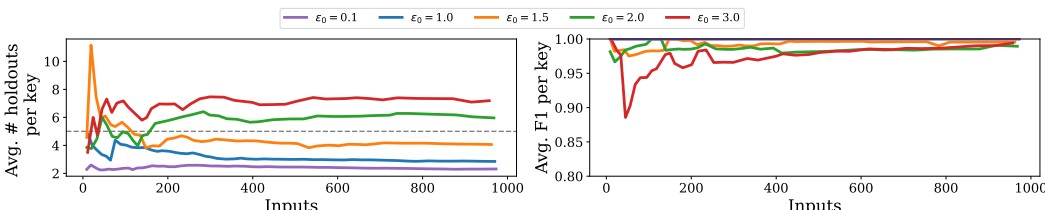

Figure 7: Interpreting generalizability of GRACE codebooks. F1 is computed on unseen holdout edits that land inside any key's $\epsilon$ ball. The gray dashed line denotes the true average number of rephrasings per observed edit.

## E   Parameter Efficiency

As introduced in the main paper, GRACE's memory requirements are small. This is because a new edit only requires $|h^{l-1}| + |h^l| + 1$ parameters, where $|h^{l-1}|$ is the dimension of the key, $|h^l|$ is the dimension of the value, and $\epsilon_{\text{init}}$ is a scalar. Further, the key's $|h^{l-1}|$ parameters are frozen and do not require learning. At inference time, activating the Adaptor only alters the model's predictions according to the $|h^{l-1}| + |h^l| + 1$ parameters of the chosen entry. As a concrete example, consider a GRACE codebook with 500 entries that edits the 60 million-parameter T5 model. For 500 edits, we would then store $500 * (|h^{l-1}| + |h^l| + 1) = 500 * (1024 + 512 + 1) = 768,500$ parameters, of which only $500 * (512 + 1) = 256,500$ are learnable. $768,500$ is only 1.3% of T5's original 60 million parameters. Edits also appear sporadically in practice, so accruing 500 edits may easily span thousands or millions of inputs. While GRACE codebooks do grow over time, adding parameters according to the data size is quite attractive in such sequential setups. Other Adaptor methods, for instance, initialize all their weights early on, leading to a far harder learning problem. MEND, for instance, predicts a 1024-dimensional and 512-dimensional vector, requiring 1,310,720 new parameters to be learned all at once. This is reasonable in their setup with training edits, but is not directly applicable to our setup. Defer uses a comparable parameter count to GRACE, adding 525,312. Memory, on the other hand, is tiny, adding just 26,624 learnable parameters because we set $N = 50$. Early experiments with Memory indicate that increasing $N$ is unhelpful.

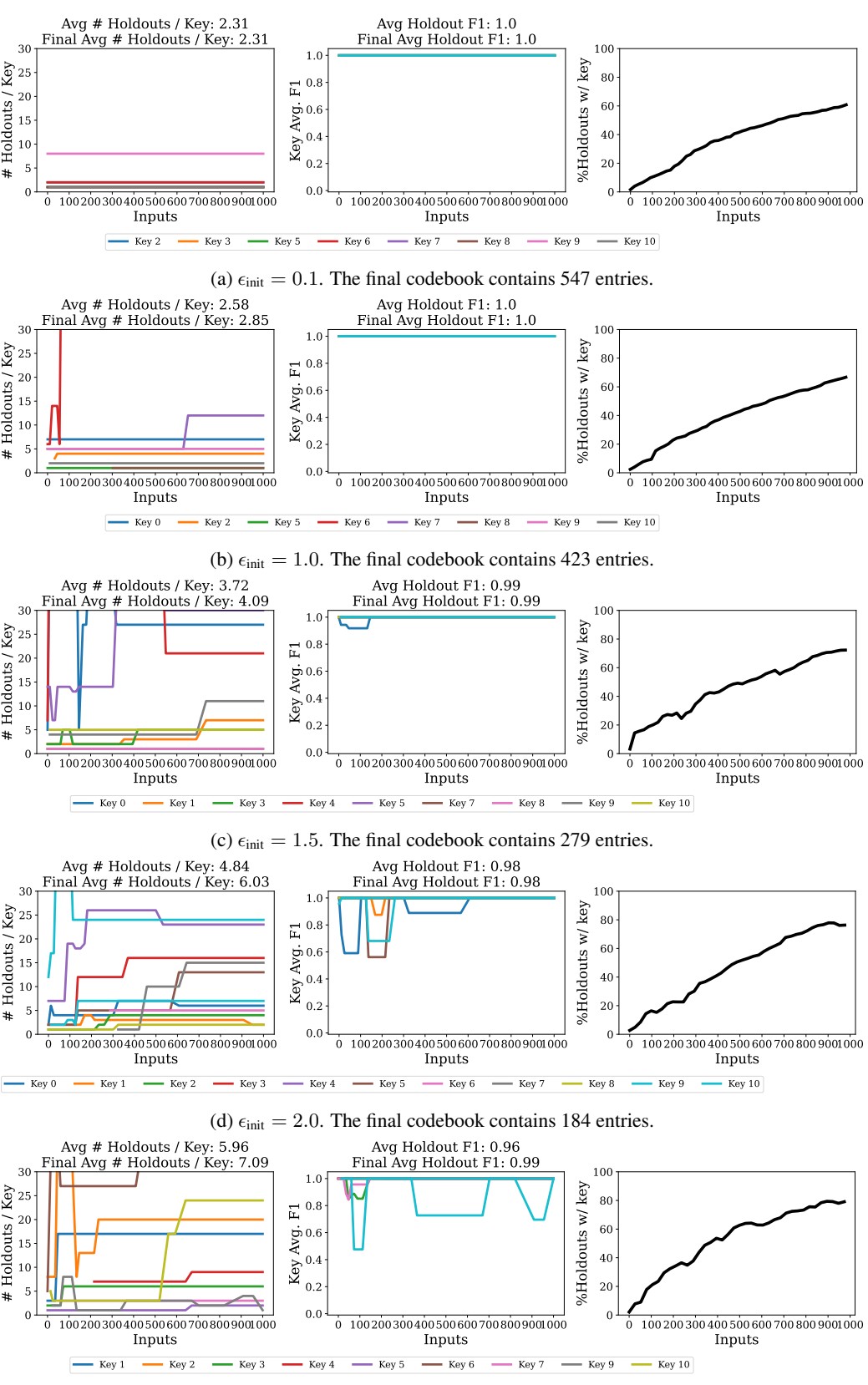

(a) $\epsilon_{init} = 0.1$. The final codebook contains 547 entries.

(b) $\epsilon_{init} = 1.0$. The final codebook contains 423 entries.

(c) $\epsilon_{init} = 1.5$. The final codebook contains 279 entries.

(d) $\epsilon_{init} = 2.0$. The final codebook contains 184 entries.

(e) $\epsilon_{init} = 3.0$. The final codebook contains 150 entries.

Figure 8: Interpreting per-key codebook behavior for 1,000 inputs for Block 4 of T5. At each step of editing, we record how many holdout samples are captured by a key and if so, which key. We also report the percentage of the holdout dataset that is captured by *any* key in the right-hand panel.

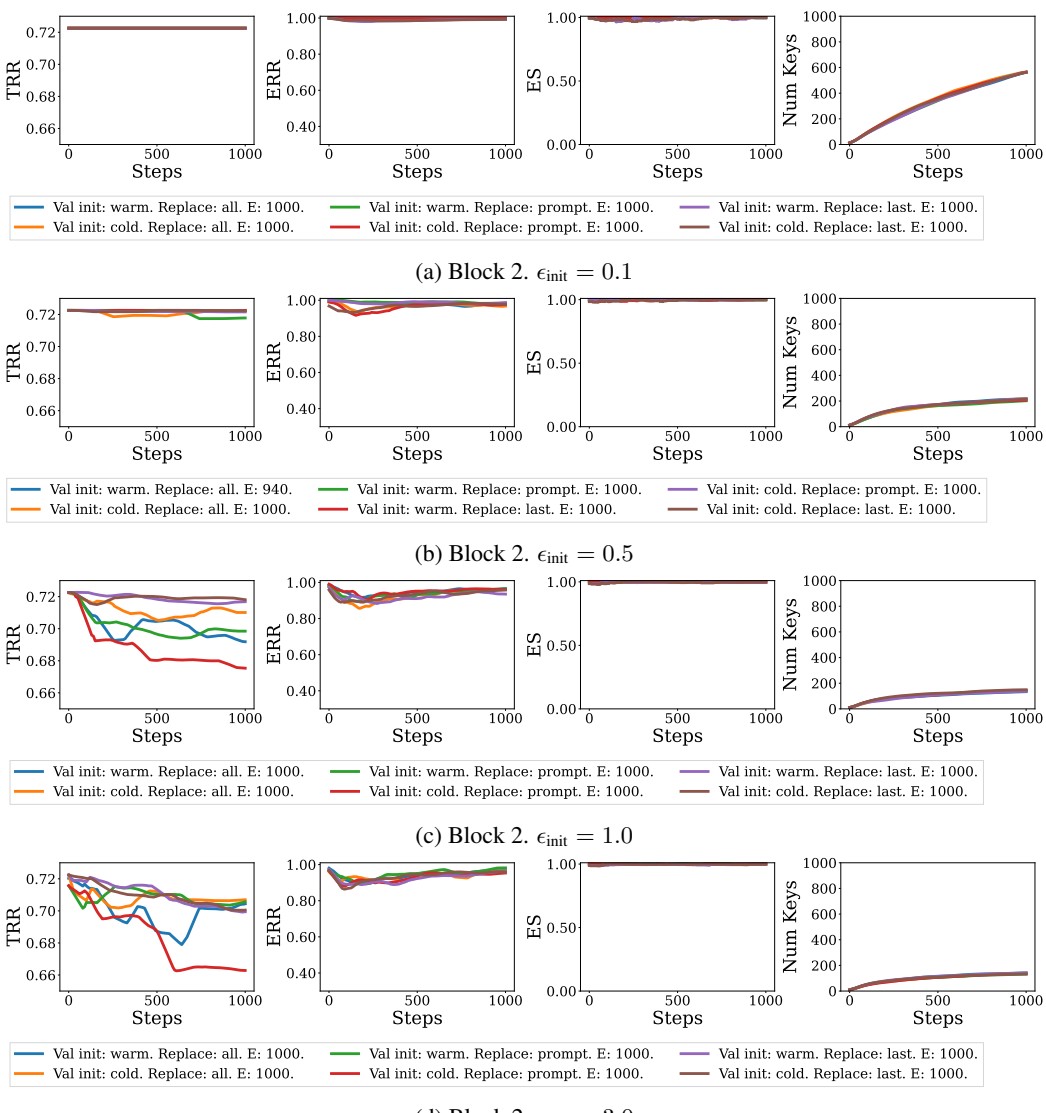

Figure 9: Ablation Study on editing T5 for zsRE on 1000 inputs. "E" denotes the number of edits performed. "Steps" denotes the candidate edit in the edit set, since over 1000 inputs, not all will be flagged as errors.

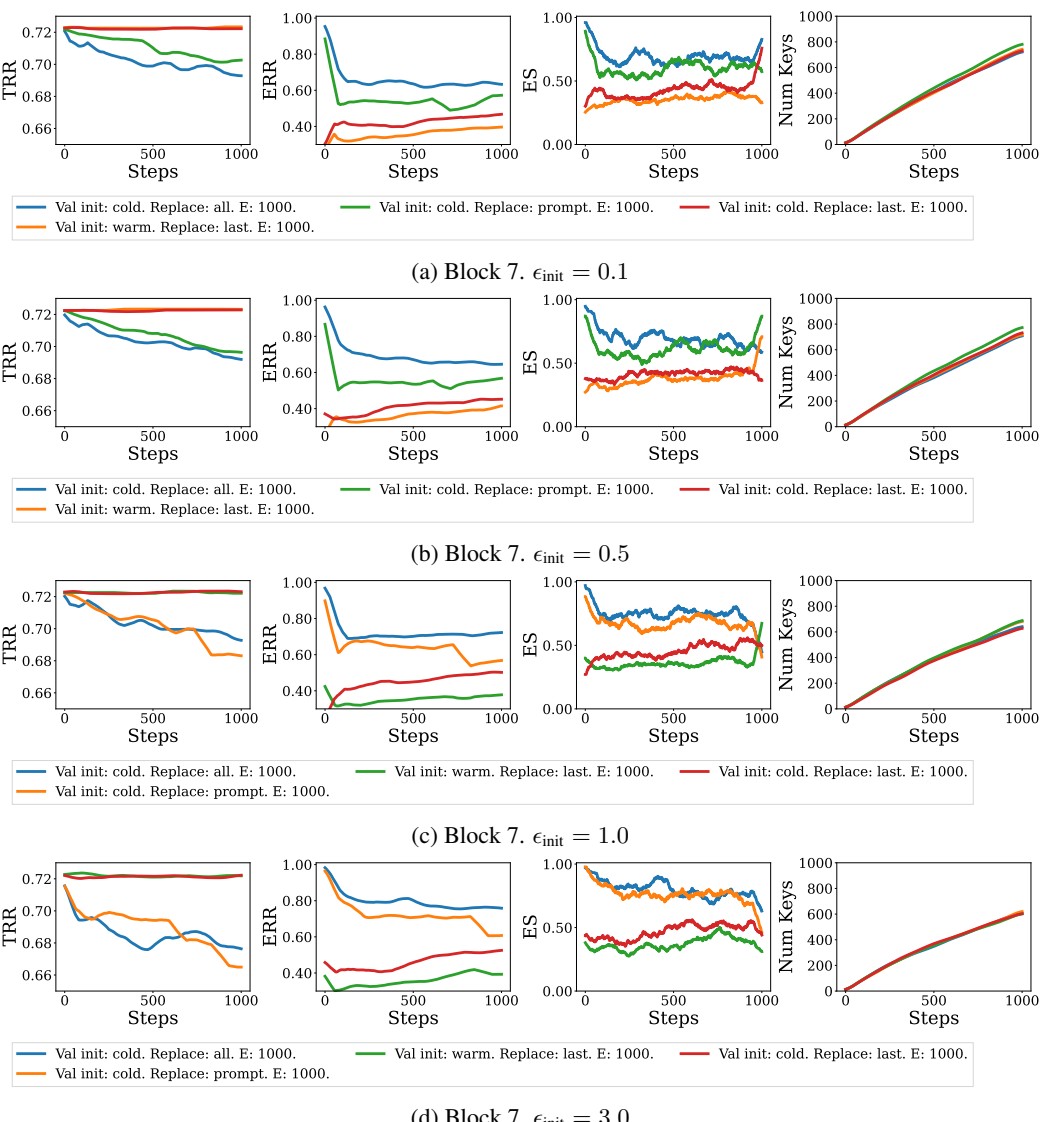

(a) Block 7. $\epsilon_{\text{init}} = 0.1$

(b) Block 7. $\epsilon_{\text{init}} = 0.5$

(c) Block 7. $\epsilon_{\text{init}} = 1.0$

(d) Block 7. $\epsilon_{\text{init}} = 3.0$

Figure 10: Ablation Study on editing T5 for zsRE on 1000 inputs. "E" denotes the number of edits performed. "Steps" denotes the candidate edit in the edit set, since over 1000 inputs, not all will be flagged as errors.

| GRACE KEY | EDITS | TRUE LABEL |
|---|---|---|
| 0 | What type of voice is Gemma Bosini?
What kind of voice is Gemma Bosini?
What is the voice of Gemma Bosini? | soprano |
| 7 | In what war did Carlos W. Colby fight?
What war was Carlos W. Colby fighting in?
What war or fight did Carlos W. Colby fight in?
What war did Carlos W. Colby fight in?
What war or battle was Carlos W. Colby fighting in? | American Civil War |
| 8 | By whom was the Queen Victoria Building designed?
Who was the Queen Victoria Building designed for?
Who was Queen Victoria Building designed by?
Which person designed Queen Victoria Building?
Which architect designed Queen Victoria Building? | George McRae |

Table 4: Examples of GRACE codebook entries. "GRACE Key" is the index of the chosen codebook entry for these inputs. "Edits" are inputs that were wrong prior to editing and are fixed by the same key. "True Label" is the true label associated with these inputs and the codebook's learned value.

# F   Ablation Study

We ablate GRACE components by comparing four design choices using different layers and choices of $\epsilon_{init}$ using T5 on up to 1000 edits from zsRE. The hyperparameters we search across are shown in Table 3. We focus this study on the T5 encoder's Blocks 2 and 7, which pose significantly different behavior.

As our results in Figure 9 show, there are interesting relationships between value initialization and token replacement. Starting with the right-most panels, we excitingly observe that GRACE achieves significant edit compression: For up to 1000 edits, the GRACE codebooks often end up with only about 200 keys. This marks success in our codebook maintenance strategy, since this can only be achieved if $\epsilon$ values are increasing appropriately for new edits. For small $\epsilon_{init}$ values, the number of keys grows larger, up to about 600, demonstrating the trade-off between codebook size and memorization. This trade-off is seen in the three left panels measuring Training Retention Rate (TRR), Edit Retention Rate (ERR), and Edit Success (ES), all in terms of F1. For the smallest $\epsilon_{init} = 0.1$, we see these three metrics remain high throughout editing. However, looking down through Figures 9a, 9b, 9c, and 9d, we see TRR and ERR drop as the codebook generally trades size for generalizability. This trend remains true when editing Block 7 (Figure 10), though this later layer appears far harder to edit successfully. This may hint at a failure mode of GRACE that may be addressed by further hyperparameter tuning: If a new key is initialized yet fails to correct the model's behavior, even identical future inputs will still require edits. This may be addressed by training values for longer or adding new value training methods: In principle, learning a new GRACE value is no different from training any model, so may benefit from any advanced optimization method.

For hyperparameter `Replacement Token`, we see that replacing just the value for the *Last* input token appears to be superior when editing Block 2, especially for TRR. This makes sense because replacing only the last input token will likely have the smallest impact on unrelated inputs. Excitingly, this replacement strategy remains on par with the others for ERR and ES. For Block 7, however, the opposite is true. Given the low overall ES, it remains unclear how replacement strategies differ in general for this layer.

For hyperparameter `Value Initialization`, we see negligible differences between *warm* and *cold* starts for Block 2. For Block 7, *cold* appears overall worse than *warm*, despite having higher ES.

**Edit Compression.** Compressing multiple edits into singular keys is bounded by the number of unique desired labels since each GRACE value points to one desired output. For example, if new edits come from 5 different classes, GRACE must include at least 5 keys. As we saw in Figure 9, GRACE can compress many edits into single codebook entries successfully. Table 4 shows 3 examples of inputs that all lead to the same key.

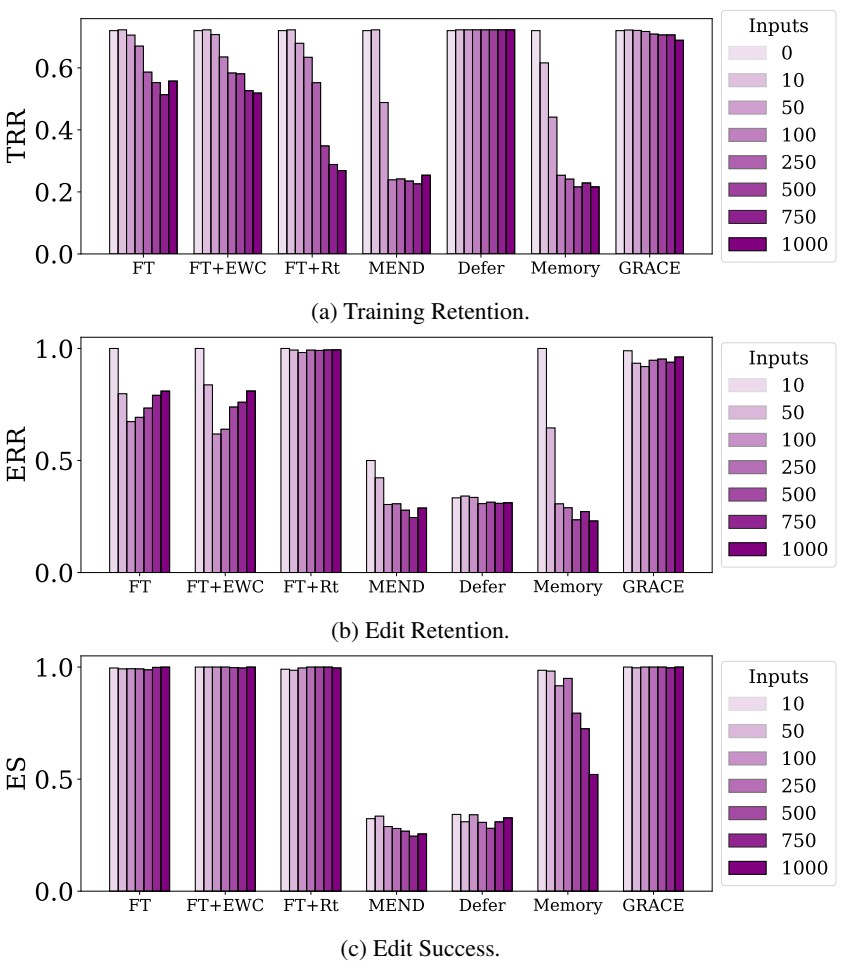

(a) Training Retention.

(b) Edit Retention.

(c) Edit Success.

Figure 11: Comparing editors over time when editing T5 on zsRE.

## G  Extended Main Results

To supplement Table 1 in the main paper, we include all metrics computed throughout editing for all comparisons. We first show barplots in Figures 11, 12, and 13. We see that the trends match the main paper: GRACE achieves strong performance compared to the comparisons. We also show finer-grained versions of these plots computed for each edit over time in Figures 14, 15, and 16.

## H  Extended Hyperparameter Study

To further extend our study on the impacts of the most important hyperparameters $\epsilon_{\text{init}}$ and the chosen layer, we run a far-longer experiment on the QA editing task, making up to 5,000 edits. Our results from this experiment are shown in Figure 17. In this finer-grained analysis, we find that our main claims remain true: some layers are better to edit than others, GRACE succeeds to generalize to previously-unseen inputs, $\epsilon$ controls the trade-off between memorization and generalization, and GRACE codebooks stabilize in size over time. Performance remains the same above $\epsilon = 6.0$. Interestingly, for $\epsilon = 2.0$, Layer 6 appears to be highly unstable over time.

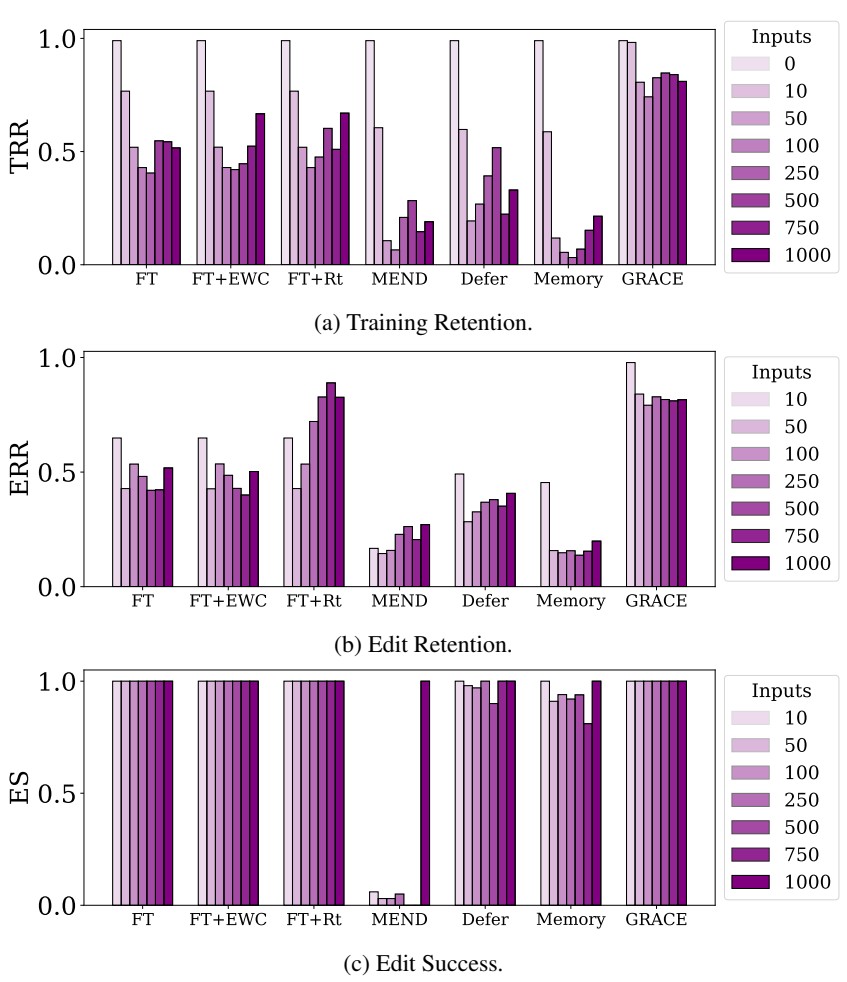

(a) Training Retention.

(b) Edit Retention.

(c) Edit Success.

Figure 12: Comparing editors over time when editing BERT on SCOTUS.

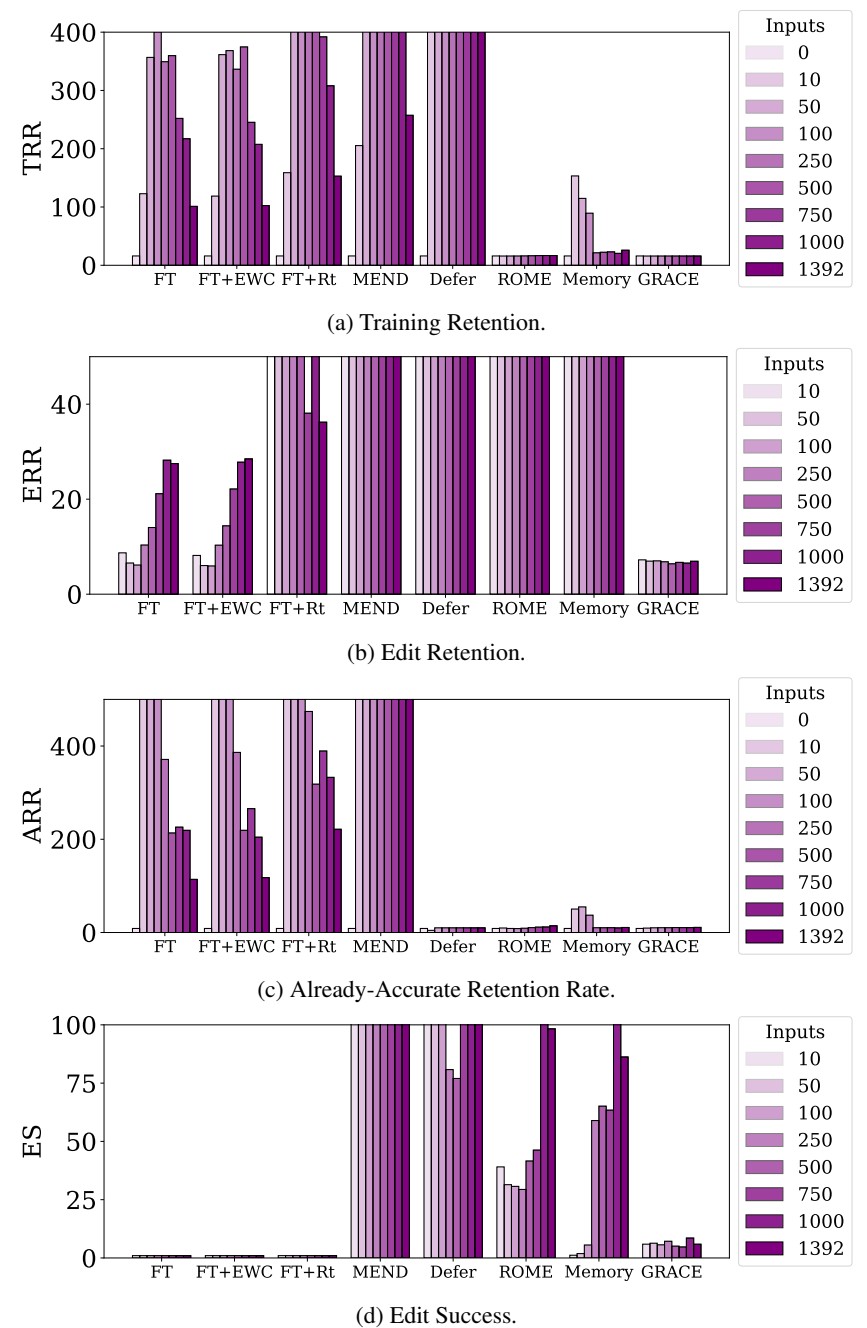

(a) Training Retention.

(b) Edit Retention.

(c) Already-Accurate Retention Rate.

(d) Edit Success.

Figure 13: Comparing editors over time when editing GPT2-XL on Hallucination.

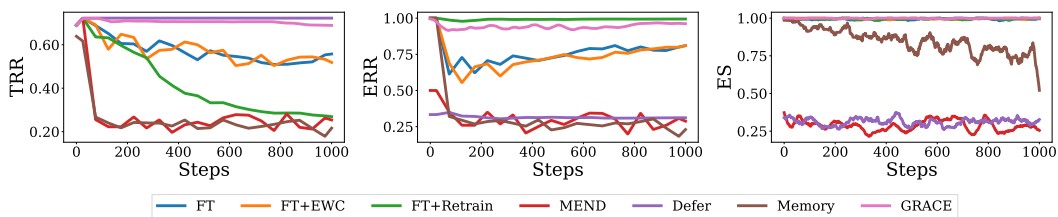

Figure 14: Comparing all methods throughout editing on zsRE.

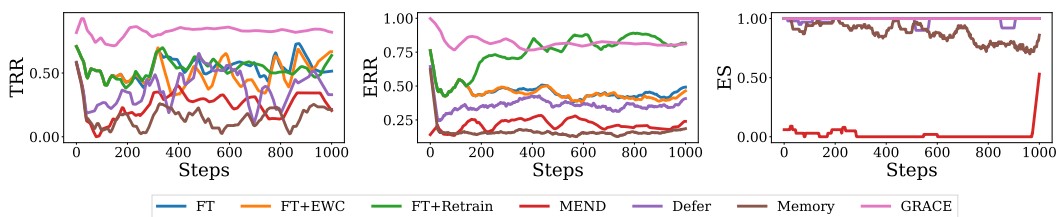

Figure 15: Comparing all methods throughout editing on SCOTUS.

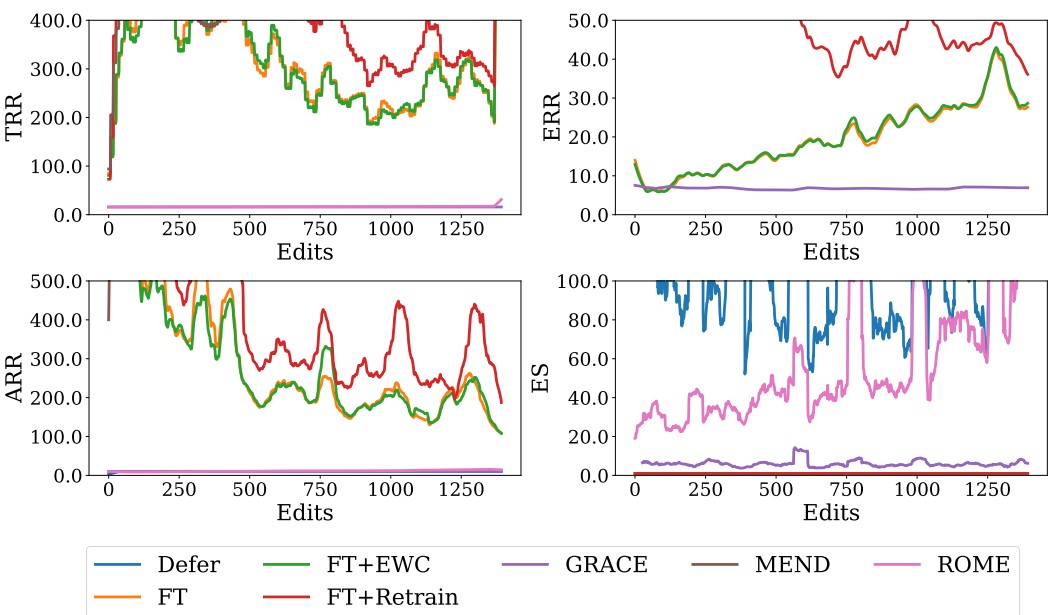

Figure 16: Comparing all methods throughout editing on Hallucination.

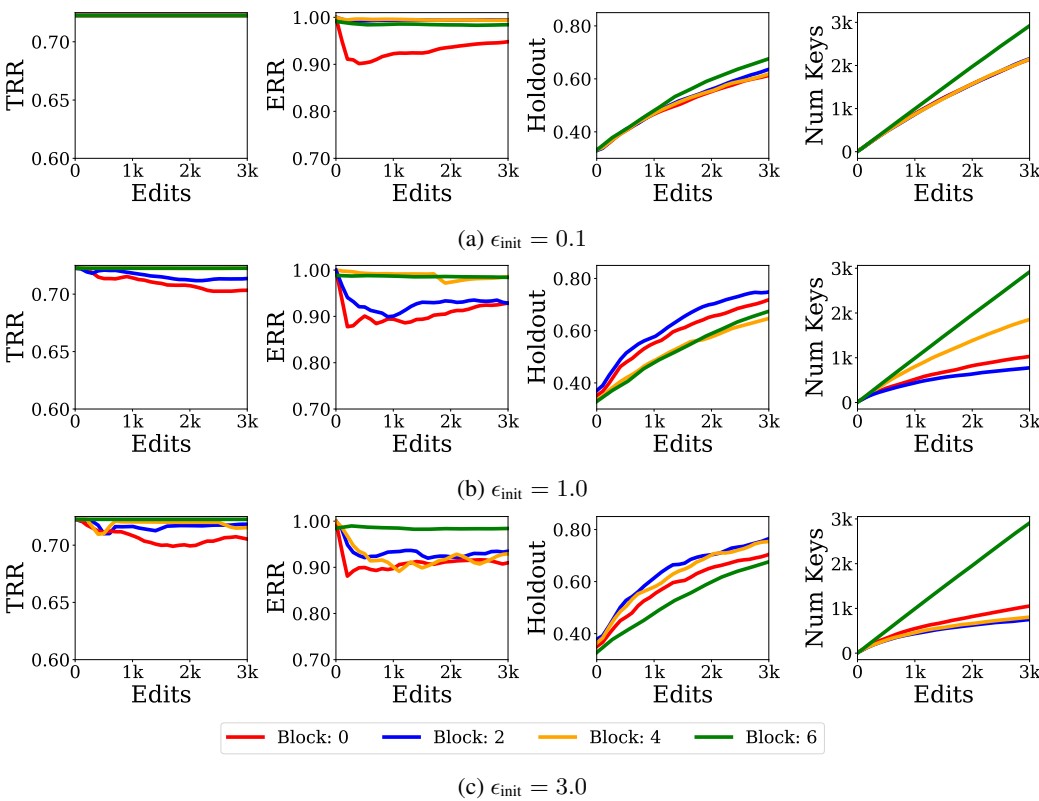

(a) $\epsilon_{\text{init}} = 0.1$

(b) $\epsilon_{\text{init}} = 1.0$

(c) $\epsilon_{\text{init}} = 3.0$

Figure 17: Impact of $\epsilon_{\text{init}}$ and block choice for GRACE editing T5 on zsRE for 3000 sequential edits. Along with TRR and ERR, we also measure F1 on a "Holdout" edit set containing unseen rephrasings of all 3k edits. We find that editing blocks 0 and 6 use more keys and achieve higher TRR, but lead to lower ERR and generalize worse, as shown by lower "Holdout" values.

