# OpenReview forum: "Aging with GRACE: Lifelong Model Editing with Discrete Key-Value Adaptors"
_NeurIPS.cc/2023/Conference — NeurIPS 2023 poster_

### Official Review · Reviewer_2Vyt · 2023-07-08

**Soundness:** 2 fair
**Presentation:** 3 good
**Contribution:** 2 fair
**Rating:** 4
**Confidence:** 4

**Summary:**

The paper suggests a new method for continual model editing. To deploy a parameter-efficient model editing remedy, the authors introduce an adapter for a target layer, keeping the pre-trained model weights intact yet revising the model prediction correctly. The suggested adapter contains a codebook - key-value pairs and corresponding distance coefficient \epsilon. By training them via a fine-tuning mechanism, they can edit the original prediction accordingly.

**Strengths:**

The proposed method introduces the interesting idea of continual model editing. The basic idea of codebook updating and introducing new entries is simple yet reasonable. And suggested method outperforms baselines by a significant margin. And the ablation study done in the supplementary file is also good.

**Weaknesses:**

The most critical limitation is that the choice of editing layer is essential, but it is achieved via heuristic search. Recently released deep neural network architectures provide a depth (layerwise/blockwise) variety according to the scale, and heuristic search to find the best layer to edit is a nontrivial problem. Unfortunately, as shown in the analyses and the limitation section of the main paper, this limitation is the main obstacle to the practical usability of the suggested model.

Also, most baselines are not targeted to the continual learning setting, so the benefit of the suggested method for continual model editing is a bit difficult to measure fairly. Simple combinations of existing (and recent) continual learning methods with model editing baselines can be better competitors to reveal the merits of GRACE more intuitively. And it will be also great to provide not only model performance like TRR, ERR, or ARR, and the change of additional memory overhead over long model edit sequences.




**Questions:**

N/A

**Limitations:**

Please see the weakness.

---

> ### Author Rebuttal · Authors · 2023-08-09
>
> Thank you for your feedback, we are glad for the opportunity to clarify some points.
>
> **Weakness 1: Sensitivity to layer selection**. As shown in our Updated Results PDF, GRACE is far less sensitive to layer choice than we originally found. This implies picking hyperparameters before deployment may not be dealbreaking, especially in heavily-constrained deployment settings.
>
> Towards practical use, even though GRACE is more robust than we initially reported, hyperparameter selection in any online setting is hard. Our exact problem setting is particularly restrictive and attractive, but indeed will benefit from growing community interest. And while tuning in our precise setting can't be done using conventional methods (e.g., cross validation), there do exist some avenues towards tuning in practice. For instance, in some settings it may be possible to collect a small dataset beforehand and tune our (few) hyperparameters. For instance, there could be exogenous data (like ROME uses Wikitext) or internal validation sets to pick the layer and $\epsilon_\text{init}$. And in some cases, there may be ways to reset or update GRACE codebooks periodically. In fact, one strength of GRACE-style edits is that the model weights are frozen, so edits can be reverted or updated without requiring model checkpoints.
>
> **Weakness 2: Continual Learning comparisons**. While extending existing methods into our setting is possible in principle, it is far from trivial. Still, we do adapt each baseline into a continual learning setup. For Finetuning, we use existing continual learning methods (EWC and periodic retraining). For the others, we adopt continual finetuning. While they indeed underperform, we believe that further adaptation strays from their original proposals and too close to sheer model development, given our understudied problem setting. We hope that future works will study this setting further and surpass GRACE, maybe by adapting these existing methods in intelligent ways. Figure 1 in the Updated Results PDF includes one notion of "memory overhead over long model edit sequences", as measured by codebook size (Num Keys). After each edit in the sequence, we compute and report the number of keys in the codebook for different choices of $\epsilon_\text{init}$ and layer. To further address your comment, we will compute and report the actual amount of memory required to store the codebook in the Appendix. We also believe that making GRACE-style editing faster and more memory-efficient is a worthy research direction that will benefit from community focus.

---

> > ### Author Response · Authors · 2023-08-18
> > **Kind reminder of our response**
> >
> > Thank you again for your feedback on our work! We'd like to kindly remind you that we've addressed your concerns. Since there's very little time remaining for the discussion period, we'd love to know if you have any lingering concerns that we can address within the timeframe.

---

> ### Comment · Area_Chair_Ynxz · 2023-08-19
> **Gentle Reminder by AC**
>
> Dear Reviewer,
>
> Could you carefully read the authors' rebuttal as well as the others' reviews and their rebuttal, and make responses at your earliest convenience? The deadline for the discussion phase is fast approaching, which is due Aug 21st 1pm EDT, so your quick responses will be greatly appreciated.
>
> Best,
>
> AC

---

### Official Review · Reviewer_wTwG · 2023-07-08

**Soundness:** 3 good
**Presentation:** 3 good
**Contribution:** 3 good
**Rating:** 7
**Confidence:** 4

**Summary:**

The authors propose GRACE, an approach to model editing that is tailored to many-edit scenarios. GRACE augments a hidden layer of a neural network with a key-value memory that overrides the output representations of the layer when the input representation falls near the stored key. The method is simple and computationally efficient, and compares favorably with existing editors. Closer analysis shows that GRACE ultimately provides a tradeoff between forward generalization of past edits and retention of pre-existing knowledge.

**Strengths:**

The authors study a clearly-defined and very relevant problem, sequential model editing, which extends the setting of applying one or a small number of model edits to hundreds or thousands, which is much more realistic. The method they describe is simple and novel, and does not include the requirements for training data that some existing methods have. GRACE also leverages the internal representations of the pre-trained model itself to (in theory) enable generalization of the edit information to related inputs (however empirical validation of this capability is mixed). The analysis of the method's memorization and generalization in 4.2.2 is welcome.

While the final results show that there is still a strong tradeoff between the ability to generalize edits forward (i.e., to future queries that ask about similar information to those queries that were already edited) and the retention of pre-existing information, I think the paper is a useful exploration of a straightforward and intuitive approach to sequential editing.

The authors also include a clear discussion of the limitations of their work, which is welcome.

**Weaknesses:**

The main weakness I see in the paper is the fact that GRACE might do little more than memorize the edits it is exposed to. The analysis in 4.2.2 (which is very welcome!) suggests that GRACE essentially has two regimes, one where we essentially memorize past edits and retain all past knowledge, but are not able to generalize edit information to future, related inputs (editing layers 4 or 6 in the $\epsilon=1.0$ plot) or where we are able to generalize edits to future, related edits, but suffer a (potentially dealbreaking) penalty in terms of the information we retain from the original task. In my view, the tradeoff as shown in the paper is not viable for real-world use cases. However, I still think the paper makes a useful and interesting contribution to model editing/sequential model editing.

One other point about related work:

I think L79 is wrong or misleading; MEND and ROME are only loosely based on PEFT. MEND only requires a single fine-tuning step, and neither ROME nor MEND is more prone to overfit than regular fine-tuning to my knowledge. Whether or not PEFT overfits more than regular fine-tuning isn’t relevant here, IMO.

**Questions:**

L156: Does this expansion process ignore the fact that we can have multiple word senses? For example, if we have an edit for “the robber of the ___” and “the house by the river ___”, even though the labels for both edits might be “bank”, it seems like we’d want to represent these separately.

L257: “unsurprisingly, MEND underperforms since it forecast edits for new, unobserved inputs” What does this mean?

L87: what privileged information does MEMIT use?

L139: Can we not have the case where the hidden state is not within the deferral radius of the closest GRACE key, but the hidden state is within the deferral radius for the n-th closest key? Considering only the closest key seems like it ignores this possibility.

What, in the authors' view, is the primary cause of the difficult tradeoff between retention of old information and forward generalization of edits? Currently, GRACE either requires a problematic destruction of pre-trained knowledge or a failure to apply edits to future, related edits.

**Limitations:**

The authors include a clear discussion of relevant limitations of their work.

---

> ### Author Rebuttal · Authors · 2023-08-09
>
> Thanks for your valuable and positive feedback, we address your hesitations below:
>
> **Weakness 1: Memorization vs. Generalization**. We are happy to report that the new results included in the Updated Results PDF above address this weakness: GRACE layers can generalize far better than previously reported, leading to better trade-offs. The trade-offs still exist, but we find there is room for better trade-offs between retaining training information while making generalizable edits. At the very least, we do not believe the penalty remains dealbreaking, though we agree this is a crucial point and we look forward to future works improving in this direction.
>
> **Weakness 2: Referencing "PEFT"**. Thanks for catching this, we agree and will omit the direct use of “PEFT.” Instead, we will simply call them “parameter efficient” and omit the sentence about overfitting.
>
> **Question 1: Multiple word senses**. The expansion process should not ignore multiple word senses because the model’s representations are contextual. In this example, “the robber of the” and “the house by the river” are likely different representations, so expanding an edit from one should not impact the behavior of another. However, we don’t account for this explicitly, so in the case that the representations are quite close, there may be interference. Settings where the context doesn’t separate representations for sentences like this pose an interesting avenue for future investigation. On the value-learning side, GRACE actually poses a way to avoid multiple word senses when it's unnecessary. For instance, if  “the robber of the” and “the house by the river” should both point to "bank," and there's no further generation required, it may be inefficient to learn different embeddings for the same word.
>
> **Question 2: "forecasting" edits**. Here we mean that MEND trains a hypernetwork before making edits, then is used to predict weights for edits that it didn’t see during training. Further, since MEND is trained on previous versions of the model, as it edits the model, its capacity to edit the model also diminishes, because it’s being asked to edit a model it wasn’t trained to edit.
>
> **Question 3: "privileged information"**. In both ROME and MEMIT, “covariance statistics are collected using 100,000 samples of Wikitext” [[Meng et al., ICLR’23; Appendix B.4](https://arxiv.org/pdf/2210.07229.pdf)]. While we let ROME use these data in our comparison, this information is unavailable according to our problem definition. If “privileged” is the wrong word, we’d be happy to use “external” or “exogenous” or something else per your recommendation.
>
> **Question 4: Queries inside n-th closest key's deferral radius**. This is an excellent point and indeed considering only the closest key ignores this possibility. Since two nearest keys can't have overlapping deferral radii, this case might be rare given the edit would have to be between the existing keys. Changing this choice in code is trivial, so we will address this comment by investigating how the results change if we check for any overlapping deferral radii. We believe our results still demonstrate the value of our method as proposed and that this alternative solution may in fact improve performance further by making edits more generalizable. Further, the cost of searching across all keys instead of only the closest might eventually outweigh the benefits posed by addressing this possibility. Thanks for pointing this out, we’ll add a sentence noting this is possible.
>
> **Question 5: Remaining challenges**. Our Updated Results PDF demonstrates that GRACE’s trade-off is better than previously shown. We believe this addresses the core of your point. However, moving forwards, achieving a good trade-off is an important area for future work. We believe more work is needed to find high-impact regions to make edits, e.g., identify whether there are more- or less-generalizable regions in models to make edits. Specifically, if models encode semantic similarity for different topics in different places throughout the whole model, targeting these areas could provide the best trade-off for retention and forward generalization.

---

> > ### Author Response · Authors · 2023-08-18
> > **Kind reminder of our response**
> >
> > Thank you again for your feedback and for supporting our work! We'd like to kindly remind you that we've addressed your concerns. Since there's very little time remaining for the discussion period, we'd love to know if you have any lingering concerns that we can address within the timeframe.

---

> > ### Comment · Reviewer_wTwG · 2023-08-19
> >
> > I appreciate the authors' comprehensive responses to the reviewers' feedback. The TRR is significantly improved in the revised results, and it does look like GRACE is learning something beyond pure memorization.
> >
> > I still feel that more focused analysis of edit generalization on something other than rephrases is crucial to make model editing useful in the real world. Nonetheless, I think the ideas and updated evaluations in this paper are sufficient to warrant acceptance. Therefore I will raise my score to 7.

---

> ### Comment · Area_Chair_Ynxz · 2023-08-19
> **Gentle Reminder by AC**
>
> Dear Reviewer,
>
> Could you carefully read the authors' rebuttal as well as the others' reviews and their rebuttal, and make responses at your earliest convenience? The deadline for the discussion phase is fast approaching, which is due Aug 21st 1pm EDT, so your quick responses will be greatly appreciated.
>
> Best,
>
> AC

---

### Official Review · Reviewer_nd9W · 2023-07-16

**Soundness:** 3 good
**Presentation:** 4 excellent
**Contribution:** 3 good
**Rating:** 7
**Confidence:** 2

**Summary:**

The paper presents a new method for sequential model editing (GRACE). The approach works by modifying the output of a pretrained model's given layer $l$ through the use of an external key : value storage. Whenever an edit is required on input $x$, the model stores the input representation $h^{l-1}$ as key, and learns a value $v$ via gradient descent which yields the edited output $y_{edited}$. In order to enable generalization, the authors propose to map all representations within an adjustable (edit-specific) $\epsilon$ to the stored value $v$. Crucially, to enable sequential editing, the authors propose to update the codebook (Alg. 1) the following way : whenever a new edit overlaps with the $\epsilon$-ball of a previous edit, this $\epsilon$ is either shrunk to reduce the overlap between edits, or expanded if the two edits share the same desired output.

The authors evaluate their method according to three metrics, which look at whether the sequentially performed edits are memorized, and the model's ability to preserve its original input on non edited points. From the results, it is shown that the proposed method performs across several benchmarks. Finally, the authors perform an model analysis, looking at the impact of $\epsilon$ and the choice of layer to add the GRACE mechanism.

**Strengths:**

Originality :
 - The proposed is, to the best of my knowledge, a novel approach of an external discrete memory for sequential model editing
 - Grace seems like a solution which could actually be deployed in realistic edit scenarios

Quality / Clarity :
- The paper is well written and easy to follow.
- The intuition behind how the method works is well-built, and the model analysis section is interesting
- The experimental section is overall well-designed


**Weaknesses:**

1. Given that the choice of layer and value for $\epsilon$ seems to highly impact the behavior of the deployed system, it's unclear how practitioners would choose this value a priori. Given that this deployed in an online setting, standard cross-validation cannot be used directly.
2. The experimental section is somewhat limited, as the authors only look at NLP tasks. That being said, given that the authors evaluate different model scales, this is more a suggestion on how to make the paper more complete than a hard weakness.

**Questions:**

1. Could the authors please explain how GRACE layer deals with multi-token inputs with autoregressive models ? What is implied by "replace only the final token of an input prompt" ?
2. How are rephrasings generated for the results shown in Table 2 ?


**Limitations:**

limitations properly addressed.

---

> ### Author Rebuttal · Authors · 2023-08-09
>
> Thanks very much for your valuable and positive feedback! We respond to each of your concerns below.
>
> **Weakness 1: Hyperparameter tuning**. Our new results (Updated Results PDF) show that GRACE editing is far less sensitive to hyperparameter tuning than we originally thought, so picking the right parameters initially is lower-stakes (also discussed in the global response). Still, in practice, we agree tuning the hyperparameters is a challenge, present in many continual or online learning setups. While we leave richer solutions to future works, one practical method could be to use internal validation data or external data (e.g., ROME precomputes statistics about their models using 100,000 samples from the public Wikitext dataset).
>
> **Weakness 2: Extending to more modalities**. We absolutely agree that GRACE is applicable far beyond NLP tasks and we are excited to see how GRACE works on other modalities. Given that most recent works on model editing focus on NLP tasks, we believe our experiments present fair comparisons. Extending to more modalities is indeed an exciting direction and we hope our work inspires others to work on this in the future.
>
> **Question 1: Handling multi-token inputs**. When editing in an autoregressive setting, we start by passing a prompt into GPT2-XL. Then, the embedding for the final token in the editing layer becomes the query for which we compute a new entry in the GRACE codebook (or look for existing, nearby keys). By “final” token, we mean the last token in the input prompt. We then learn a new value just for the same, final token, which will be passed into the next layer. During inference, if we passed the exact same prompt in again, the final token output by the editing layer would then match the previously-added key and be replaced with the GRACE-learned value, inducing the desired response from the model.
>
> **Question 2: Source of rephrasings**. The zsRE dataset includes a set of rephrasings for each instance. This is why we focus our study of generalization on the zsRE dataset, since rephrasings are often hard to acquire in practice. This fact limits the direct applicability of other works that require rephrasings in order to make edits.

---

> > ### Comment · Reviewer_nd9W · 2023-08-15
> > **Answer to Rebuttal**
> >
> > Thank you for the rebuttal.
> >
> > After reading the other reviewers' comments and your rebuttal, I am still of the opinion that this paper meets the bar for acceptance. I am therefore keeping my score as-is.

---

### Official Review · Reviewer_2Kqs · 2023-07-20

**Soundness:** 3 good
**Presentation:** 4 excellent
**Contribution:** 2 fair
**Rating:** 6
**Confidence:** 4

**Summary:**

The authors propose a method to editing a model’s behavior, making corrections or specifying the desired output for a particular input. Such a method is broadly applicable, but is especially relevant in the context of large pre-trained models, which are costly to fine-tune or re-train, and whose data is often not publicly available. Specifically, the authors propose General Retrieval Adaptors for Continual Editing (GRACE), which store activations (keys) for past edits into a codebook; if a particular input is close enough to one of these stored keys, then the output of the layer is overridden by the key’s corresponding value. A deferral radius for each key allows grouping of similar edits and provides some generalization behavior. For experiments, the authors compare with a number of relevant model editing baselines, editing large language models (LLMs) hundreds to thousands of times, which is significantly more than prior evaluations. Empirical comparisons look promising across a variety of editing metrics.

**Strengths:**

S1. Relevance: This paper tackles a topic that is especially important given recent developments. Large models trained on massive data (e.g. language, vision) are seeing increased interest and more exposure to the general public with the releases of chat bots/search, image generators, etc. This is largely due to impressive results, but there have also been a concerning number of wrong, offensive, or misleading examples as well. As these models continue to be used by the public, some way to apply targeted patches is essential.

S2. Assumptions: GRACE has relatively simple and realistic model assumptions. In particular, unlike some prior work, it does not require access to the pre-training data; this is important, as such data is often proprietary and massive in scale, as well as a potential data privacy risk.

S3. Experiments: \
a) Realistic settings: While past methods tend to evaluate success of a single or a handful of edits, this paper evaluates on the ability to make hundreds to thousands of edits, sequentially. This is a significantly more realistic and utilitarian setup. \
b) Three text-editing settings: The authors include experiments on multiple settings (context-free question-answering on a T5, BERT classification on SCOTUS dataset, and correction of hallucinations on GPT-3 wiki bios). \
c) GRACE outperforms a good selection of competing methods on a number of editing and retention metrics, demonstrating strong empirical potential.

S4. The paper is well-written and easy to understand. The figures and illustrative examples do an excellent job conveying the key concepts.


**Weaknesses:**

W1. Editing examples vs concepts: From my understanding, the proposed method focuses on editing predictions of a model for a specific example. This contrasts with continual learning methods, which tend to focus on learning new concepts or classes. While the editing approach is good for correcting mispredictions or hallucinations, it doesn’t seem like it is capable of learning to make new predictions: for classification, this could represent new categories or new vocabulary.

W2. Scalability: The number of GRACE layers required in the codebook mostly scales linearly (technically sublinear because edits can be grouped with previous GRACE layers, but from Figure 4, the overall behavior is still close to linear—converging to 50% cache rate is still O(n)). In addition to storing an increasing number of GRACE layers, this increasingly large codebook also requires an increased number of comparisons at inference time, for *every* forward pass, even inputs that are correct and otherwise unaffected by the edits. Experiments show up to 5K edits, but real-world deployments may (e.g. chatbot search engines) may require significantly more.

W3. GRACE layers value storing vs override: My understanding is that the models being considered in this work are fully deterministic during inference. Thus, once a particular input triggers a particular GRACE layer anywhere in the model, then the retrieved activation $h^l$ should always result in the same output. This raises a couple questions: \
a) If the output from a GRACE layer deterministically maps to a particular output, then why wouldn’t we instead directly store the desired output and save the computation of the subsequent layers $l+1, …, L$? \
b) This also suggests a much simpler approach: maintain a dictionary of input-output corrections, and check if the input exists in the dictionary before running the model. The primary advantage of GRACE then would be the potential for better generalization, as matching to previous edits is done at the feature level, but this needs to be verified empirically.


Less critical concerns:
- Title: While the title is kind of cute, “Aging” is only partly relevant to the problem being addressed here. Edits are also necessary for examples for what the model gets wrong, even when it’s a “newborn”.
- I think it’s debatable whether the proposed method should be considered a “continual” or “lifelong” learning method, as it’s more or less sidestepping the catastrophic forgetting problem altogether with a roughly linearly increasing codebook of overrides (see W2). With the $\epsilon$ radius, there is some potential for forward or backward transfer, but I would classify it as minimal and relatively rudimentary. On the flip side though, I can see similar arguments being made about some expansion-based CL methods, though generally their scaling characteristics aim to be better.
- I was a little disappointed that there weren’t any experiments with large vision models, but I understand the desire for limiting scope.


Miscellaneous:
- Fig 2: “No near key” => “No nearby key”
- Equation between Lines 139-140: The top line doesn’t quite make sense. I believe the authors meant to write something along the lines of “if $d(h^{l-1}, K_{i_*}) < \epsilon_{i_*}$” instead. Also please include an equation number.
- Line 149: “one of two”?
- Line 191: missing period “.”
- Line 279: possibly a missing space after the “:”

========

Overall Assessment:

I found this paper interesting and well presented; the proposed GRACE layers appear to be a reasonable, straightforward solution to editing. However, I’m concerned about the method’s overall scalability, especially when the method in many ways is similar to keeping a dictionary of input-output overrides (e.g. prefacing the model inference call with simple if-then statements). By operating in feature space, perhaps GRACE has better generalizability than such a rudimentary approach, but there is little empirical evidence of this. Also, it still isn’t clear to me why values are activations for the next layer as opposed to just jumping straight to the output. I’m willing to raise my score if my concerns are sufficiently answered.


**Questions:**

Q1. What causes the fluctuations in ERR when applying GRACE to layer 6 with $\epsilon=1.0$ in Figure 4?

Q2. How does the inference time change with increasing numbers of edits?


**Limitations:**

The limitations section is well-written and frankly acknowledges some valid limitations and potential ethical concerns. While important, I don’t necessarily see these as blocking acceptance of the paper.

---

> ### Author Rebuttal · Authors · 2023-08-09
>
> Thank you for your feedback, we're glad for the chance to clarify some points.
>
> **W1: Editing examples vs. concepts**. There are many types of continual learning problems. Adding “new categories or new vocabulary” is often called “Task-incremental” or “Class-incremental” continual learning. Our problem is closer to “instance-incremental”, where data share their category/vocabulary, but arrive over time (though we observe only errors, unlike in regular continual learning). We believe this is a good starting point (our experiments show it’s hard), and we hope our work will inspire solutions in other interesting problems. In fact, GRACE *can* represent new categories, as long as the model can output the appropriate tokens. For example, in generative language modeling wherein output classes are verbalized to be tokens within the model's vocabulary. We think this is a further benefit of editing internal transformations (we expand on this when addressing W3).
>
> **W2: Clarifying “\% Cached Edits” vs. “Cache Rate”**. We would like to clarify that “\% Cached Edits” (Figure 4) is not a “cache rate”. Instead, it is the total number of edits that required new keys to be created. So if after 4k edits there are 2k keys, we would report 50\% Cached Edits. This is a confusing way to report our findings, so in Figure 1 of our Updated Results PDF, we plot the codebook size (Number of Keys) per step. Here we observe smaller codebooks than in the original Figure 4.
>
> **W2: Scaling to more edits**. We agree that extremely long sequences of edits can exist in practice. Given today's alternatives, we believe GRACE is a good place to start. But if we expect millions of edits, for instance, GRACE's caching may be troublesome. Still, this is out-of-scope for our work because our aim is to first develop a method for sequential editing when rapid finetuning is undesirable or impossible. Given a large, newly collected dataset with corrected edits, pre-training data, and enough computational resources, standard finetuning will likely be appropriate. Still, even if GRACE had a high cache rate, we believe GRACE can be scaled, especially given works that have scaled vector similarity search [[2](https://engineering.fb.com/2017/03/29/data-infrastructure/faiss-a-library-for-efficient-similarity-search/)] and growing interest in vector databases. Further, choosing between parametric and non-/semi-parametric methods is often based on data scale, which is not a problem unique to our work. GRACE allows the user to “interpolate” between these regimes. We agree clarity on promising settings will strengthen our paper, which we will discuss in our Appendix.
>
> **W3: Value storing vs. override**. Our evaluation is deterministic, even though T5, BERT, and GPT2-XL can all be decoded probabilistically. This is an attractive and popular feature, especially for language generation. GRACE does not impair non-deterministic use of any method since we do not alter the model’s outputs directly. For instance, a GRACE-edited model can still produce non-deterministic outputs if sampling is used during decoding. We believe leaving the model outputs intact is actually important, especially given today’s focus on probabilistic decoding. Per your questions:
>
> a) Retrieving a looked-up answer in the middle of the network is indeed feasible. But there are limitations depending on the task. For classification, returning the exact label seems quite reasonable (though batch inference with elements exiting the network at different times would require extra tweaking). For language modeling, though, generating exactly the same output each time (as storing the desired output would do) is not the end goal, and moreover, the embeddings are needed to be attended over during autoregressive decoding. So by caching embeddings, not outputs, we keep GRACE general purpose and hope to inspire further work on editable decoding.
>
> b) Caching input-output pairs is also possible, but removes the benefit of using semantic similarity according to the pre-trained model, as you mention. Fortunately, our Updated Results PDF does show improved generalization. Plus, improving our generalization (and that of other model editors) is an exciting and challenging task that we believe will inspire future work.
>
> **Less-critical concerns**.
> 1. *Title*: We agree that edits are necessary at each stage of deployment. By “aging” we want to imply post-training, not early vs. late deployment.
> 2. *Is ours a lifelong solution?* Fortunately, our Updated Results PDF address the trade-off significantly. We also absolutely agree that improving generalization is an important and hard step (and all model editing works find this is hard). We hope our method is the first of many steps in this direction for our problem. We agree that GRACE is applicable beyond NLP tasks and this is an exciting avenue for future work.
> 3. *Misc.*: Thank you for catching these typos, we have updated the paper.
>
> **Q1: Fluctuations**. This is a good observation. We'd like to point out that fluctuations decreased in our Updated Results PDF, and those we observe change at a smaller scale. These fluctuations imply that editing the model influenced its behavior on previous edits or training data. This is probably due to miscalibrated deferral radii, which could be why fluctuations get bigger as $\epsilon_\text{init}$ increases.
>
> **Q2: Inference Time**. We observe small changes in inference time as the number of edits increases (results below). Extreme settings (millions of edits) probably require new considerations.
>
> The following table shows the time it takes to add an edit (which includes inference) while editing GPT2-XL. The edit time barely increases as codebook size grows (experiment done on an NVIDIA A100 GPU with 40GB VRAM).
>
>  | Codebook Size | Edit Time (s) |
>  | ------------------- | -------------- |
>  | 10 | 5.76 |
>  | 100 | 6.54  |
>  | 250 | 8.10 |
>  | 500 | 5.94|
>  | 750 | 6.06|
>  | 1000 | 8.58|
>  | 1340 | 8.10 |

---

> ### Comment · Reviewer_2Kqs · 2023-08-14
> **Rebuttal Acknowledged**
>
> I thank the reviewers for the responses. I read through the rebuttals as well as the other reviews, from which I see pretty good alignment with my original review. Reviewer wTwG shared my concern on GRACE’s memorization vs generalization, Reviewer Qd7q w.r.t. comparisons with continual learning, and Reviewer nd9W on experiments being limited to NLP tasks. Many of my initial concerns/questions were mostly addressed. As a result, I raise my score to 6. Some additional comments below:
>
> Inference time: I appreciate the table on edit times as a function of codebook size, which was quite informative, but it’s not quite what I was asking about. My question instead was how does the runtime for forward-pass inference (not making an edit) change with respect to code size? Essentially my question can be answered by adding another column with “Inference Time (s)”. My concern is that “fixing”/”editing” a model’s response to a what is collectively a tiny region of the model’s input space can lead to a disproportionate increase in running time.
>
> Lifelong solution: This reply wasn’t quite aligned with my original comment either. My original concern was the question of whether saving input-output overrides (in feature space or not) really qualifies as continual learning, philosophically. I do agree there are some commonalities, but not as much as the paper’s title and discussion seem to suggest. I also agree with Reviewer Qd7q that the comparison with EWC is not quite appropriate.

---

> > ### Author Response · Authors · 2023-08-16
> >
> > Thank you for your increased support, we're glad that our responses were helpful! Since there's not much time left for author--reviewer discussion, we plan to address your lingering concerns with updates to the paper and future experimental plans, as described below along with a preliminary experiment:
> >
> > **Inference Time**: Thanks for clarifying this point, we believe we now understand your question better. To answer it, we ran the following extra experiment that we'll extend and add to the paper.
> >
> > *Experimental setup*:
> > * We made 3k edits to the QA T5 model's 4th layer with a small $\epsilon_\text{init}$ (the final codebook contained about 2k keys)
> > * Before each edit, we timed the one forward pass through both the original (unedited) model, and the edited model on one singular input (batch size of 1).
> > * We report and compare the inference time for the unedited and the edited models over the sequence of edits.
> >
> > *Caveats*:
> > * **Our implementation is not optimized for inference time**, so any results may not be reflective of practical considerations. Inference speed will likely be improved by incorporating more advanced techniques like approximate similarity search.
> > * Reporting inference for one input in isolation is not a complete picture. We'll replicate this experiment multiple times and show confidence intervals as a line graph.
> >
> > *Results and Discussion*:
> > * **Inference with the unedited model took .022 seconds on average and remained stable throughout editing, as expected**.
> > * **Inference with the edited model started at .022 seconds prior to editing and increased to .078 seconds after 3k edits (2060 keys)**.
> > * While slow-down is often acceptable depending on the importance of the edits, we do see some slow-down. But we believe it is reasonable given it's not our focus and this is already more edits than prior works. For example, assuming these single-inference numbers are accurate and increase linearly, inference would slow to 1 second only once the codebook contains 35,977 keys.
> > * A richer comparison would involve other editors, some of which require significant pre-computation and then underperform on this setting.
> >
> >
> >  | Num Inputs | Codebook Size | Unedited Model Inference (seconds) | Edited Model Inference (seconds)
> >  | :--------------: | :--------------: |  :--------------: |  :--------------: |
> >  | 0 | 0 | .022 | .022 |
> >  | 1 | 1 | .022 | .030 |
> >  | 50 | 50 | .022 | .028 |
> >  | 100 | 99 | .024 | .028 |
> >  | 1000 | 843 | .020 | .046 |
> >  | 2000 | 1518 | .018 | .063 |
> >  | 3000 | 2060 | .024 | .078 |
> >
> > **Lifelong solution**: Thanks for the extra details on this point, too, we will revisit this and circulate for feedback.

---

> ### Comment · Reviewer_2Kqs · 2023-08-17
>
> I thank the authors for the additional experiment. While it's true that some sort of slowdown is to be expected, I somewhat disagree with the authors on how acceptable the longer inference time is. 3.5x longer inference time after 3K edits is a pretty serious issue, and may not be practical in many use cases; to re-emphasize a comment in my initial review, this slowdown applies *every* time we want to use the model, as we don't know ahead of time if the input corresponds to a model edit or not, even if the portion of input space the edits apply to is small. The authors are correct that smarter similarity search techniques should partially alleviate this problem, but this reinforces my general concern about the method's scalability nonetheless.
>
> I still believe there are good contributions here, enough for warranting acceptance, but the method's scalability appears to be a clear direction for future work.

---

> > ### Author Response · Authors · 2023-08-17
> >
> > Thanks for the extra points, we wholeheartedly agree scalability is a clear direction for future work. When we add the extended version of this experiment to the paper, we will clearly state this is a limitation. Thanks for suggesting this experiment! It will directly improve the impact of our work and shine more light on where there's room for improvement.

---

### Official Review · Reviewer_Qd7q · 2023-07-26

**Soundness:** 3 good
**Presentation:** 3 good
**Contribution:** 3 good
**Rating:** 7
**Confidence:** 3

**Summary:**

The paper proposes GRACE, a method towards lifelong editing of models through the use of a codebook, a mechanism to write to/make use of the codebook without perturbing model weights. In essence GRACE can be viewed as an inserted adapter layer, which caches activations as keys, fine tunes values that the keys map to, and employs a so called deferral radius per key to decide when to activate GRACE.

----
Rating raised post-rebuttal from 6 to 7, as weaknesses were at least in parts clarified or addressed.

**Strengths:**

The paper has many strengths, starting from a well-motivated introduction and scope that tackles a highly important problem.

In general, the exposition of the paper is clear and it is well written and structured. The mechanisms are mostly explained in sufficient detailed and GRACE itself becomes intuitive through the added illustrations.

The empirical investigation further investigates a set of metrics and includes a set of meaningful benchmarks. Within the given set of explored alternatives, GRACE can be viewed to provide consistent advantages, making GRACE a good contender for this kind of problem setting.

The addition of a limitations section with some statements on potential (harmful) social impact is also appreciated.

**Weaknesses:**

The paper seems to have two primary weaknesses.

One lies in the exposition with respect to related work and its empirical comparison, the other is a potential concern with GRACE’s practical hyper parameters.

With regard to related work, the empirical comparison to continual learning works seems interesting and it is not fully clear why it is meaningful. If one were to take a look at one of the plethora of continual learning reviews (take Hadsell et al, Mundt et al, de Lange et al), they typically categorize techniques into rehearsal (memory buffers), adding components to the architecture or regularizing parameters. The comparison in this paper primarily seems to compare to fine tuning and EWC, which fall into the regularization category. Yet, to the best of my understanding, GRACE implies the use of extra parameters and memory, which would in that sense be much fairer to compare to continual learning methods to do something similar. There are many of those, but think of storing attention masks, adaptively learned gates, or memory buffers that similarly store activations. It is presently not clear how GRACE differs conceptually from these prior arts (mentioned in e.g. the above surveys) and following in this line, what the “cost” of GRACE is with respect to growth/memory of the codebook.

As mentioned above, the second concern lies in the choice of parameters, specifically how to pick appropriate values for epsilon and how to decide which layer to edit. From figure 4 it seem the system is not very robust to this choice necessarily and it becomes a trade-off with respect to what value is chosen for which layer. Whereas this may be clear, it will be important to clarify how to pick suitable hyper-parameters without being able to look into the future in lifelong editing (i.e. plotting figure 4 and deciding on how the curve develops on the x-axis); also see below question.

**Questions:**

It is not really clear how to choose GRACE’s hyper-parameter epsilon initially. The intuition has been nicely explained, i.e. large values will apply edits with more influence, but it is not obvious how one can pick the value in advance before starting to do lifelong editing for an arbitrary dataset. In this scenario, hyper-parameter search does not seem possible because one cannot look into the future. Additionally, figure 4 also shows even more trade-offs with respect to choice of layer, which seems to be a second hyper-parameter in that sense. It is not clear how to decide on what layer to edit at which value, specifically if the values of epsilon for different layers need to be different for GRACE to work effectively in practice.  It will be appreciated to get clarifications here in the rebuttal.


**Limitations:**

Limitations are mentioned appropriately and are appreciated. Divining deeper into the mentioned limitation of slowing down inference through similarity computation, along with potential other limitations on storage cost and codebook overhead, will be a further improvement to the paper and its limitations section.

---

> ### Author Rebuttal · Authors · 2023-08-09
>
> Thank you very much for your feedback and positive response, we have addressed your hesitations below.
>
> **Weakness 1: Empirical Comparison to Continual Learning and Cost**. We agree that our setup differs from standard continual learning, but we think some comparison is appropriate. As you note, we do compare with EWC. Beyond that, we also compare against periodically re-training the model on a cache of previous edits (Finetune+Retrain), which is a form of replay. And while they are not existing continual learning methods, MEND, Defer, and Memory do add extra architecture components. Overall, our focus is on sequential model editing, so the differing setup does bar more direct comparison. Still, we believe these comparisons have some overlap with continual learning methods and shed light on potential solutions to our problem setup.
>
> The cost of GRACE in parameter count is discussed in Section 4.2.3: Adding a new edit requires $|h^{l-1}| + |h^l| + 1$ parameters, where $|h^l|$ is the dimension of the hidden state computed at layer $l$. Per the details in Appendix D, in our T5 experiments, for example, 500 edits is equivalent to only 1.3% of T5’s total number of parameters. While this indeed grows over time, we believe the benefits of editing outweigh the costs for the scales we consider. Further, our new results show that GRACE edits are more general than previously-thought, decreasing the growth-rate of our codebooks. We believe future works will continue to make even more generalizable edits and maintain even smaller codebooks.
>
> **Weakness 2 and Question: Hyperparameter selection**. We are happy to note that our Updated Results PDF shows that GRACE is more robust to hyperparameter selection than previously thought, so there is less pressure to get this choice exactly correct. Still, hyperparameter selection in continual settings is indeed a challenge. We target the hardest setting where we assume access to no edits before applying GRACE, which is desirable and warrants further study. However, sometimes it may be possible to collect a small dataset beforehand and tune our (few) hyperparameters. For instance, there may be ways to use exogenous data (like ROME uses Wikitext) or internal validation sets to pick the layer and $\epsilon_\text{init}$. Further, in some settings, there may eventually be practical ways to reset or update GRACE codebooks throughout editing. This is actually a strength of GRACE-style edits: By leaving the model’s weights unchanged, only the codebooks require updates, so model checkpoints don’t have to be stored in order to revert any edits.

---

> > ### Comment · Reviewer_Qd7q · 2023-08-14
> > **Rebuttal is acknowledged**
> >
> > I thank the authors for the rebuttal.
> > In particular, the clarifications, extra figures and discussion of hyper parameter selection is appreciated.
> >
> > I still think there can be a more suitable set of continual learning methods to be contrasted, but also acknowledge that having EWC/fine-tune is appropriate for some context. Ideally, both these simple baseline and even more related ones would exist, but I nevertheless raise my rating to accept, given the extended hyper parameter analysis and the other parts of concerns that were addressed.

---

> > > ### Author Response · Authors · 2023-08-16
> > >
> > > Thank you for your increased support, we're glad that our responses were helpful! Given limited remaining time for author--reviewer discussion, we will address your remaining concerns by stating in the paper's experiments that EWC/fine-tune are fairly simple, preliminary comparisons and our aim is to provide this context. We do hope our work will inspire future such comparisons or adaptations of more-advanced methods.

---

### Author Rebuttal · Authors · 2023-08-09

Thank you to all five reviewers for your constructive and thoughtful feedback. We are excited to see positive reception and we summarize our understanding of the paper’s strengths as follows:
* *Problem setting*: reviewers find we have a “**well-motivated introduction and scope that tackles a highly important problem**” (Qd7q) that is “**clearly-defined and very relevant**” (wTwG), while recognizing that finding “**some way to apply targeted patches is essential**” (2Kqs).
* *Method*: reviewers note how GRACE achieves “**simple and realistic model assumptions**” (2Vyt) that is matched by a “**novel approach [that] could actually be deployed in realistic edit scenarios**” (nd9W), importantly “**does not require access to the pre-training data**” (2Kqs), and remains “**simple and computationally efficient**” (wTwG).
* *Experiments*: reviewers find that GRACE “**outperforms a good selection of competing methods on a number of editing and retention metrics, demonstrating strong empirical potential**” (2Kqs), provides “**consistent advantages**” (Qd7q), and “**outperforms baselines by a significant margin**” (2Vyt) on experiments that are “**well-designed**” (nd9W).
* *Contributions*: reviewers find our contributions significant, noting how “**the paper is a useful exploration of a straightforward and intuitive approach to sequential editing**” (wTwG), while studying a “**more realistic and utilitarian setup**” (2Kqs) than prior works.

**Updated Results PDF**. We include updated results in the attached PDF. As shown in updated Figure 1 and Table 1, we find that **GRACE editing is more generalizable and less sensitive to hyperparameters**. These results are from an extended experiment (5k edits of T5 on the QA task) that we conducted after identifying and fixing a minor bug while reproducing all model results post-submission. In these updated results, we reran comparisons on all methods. While we hurried to address your feedback this week, our experiments editing layer 6 have ~2k edits remaining at the time of this submission. We believe the findings so far will extrapolate to the remaining edits, but will update the paper with the completed experiment. **We found that our results had been deflated in our initial submission** (more keys were being created than necessary), and our findings are updated in two ways:
1. **GRACE’s generalization is far better than we initially found**—cache rate is decidedly sublinear for the QA task, which uses a much smaller codebook (137 keys for 1000 edits).
2. **GRACE is not so sensitive to hyperparameter tuning**—interior layers are similar and $\epsilon_\text{init}$ better interpolates the trade-offs. Choosing the editing layer only varies TRR within [0.7, 0.72] and ERR within [0.89, 1.0]. Choosing $\epsilon_\text{init}$ varies TRR within [0.69, 0.72] and ERR within [0.87, 1.0].

These updated findings significantly address the reviewers’ main concerns, replacing Figure 4 in the main paper, though we are excited to see further advances in future work.

**Concerns Shared by Multiple Reviewers**. Two main reservations are shared by reviewers: **Trade-offs between memorization and generalization** and **Hyperparameter tuning**. Fortunately, our updated results (discussed above) address trade-offs substantially, where we find GRACE can succeed with far-smaller codebooks. For hyperparameter tuning, we acknowledge that tuning hyperparameters in our setting is a challenge. While this challenge exists in other lifelong settings, we believe that increased community focus on our setting will spark further advances. As it stands, we see two settings where GRACE can be applied directly:
1. *Our precise problem setting*: Lifelong editing without exogenous data. Here, it’s not possible to follow standard hyperparameter tuning practices (e.g., validation sets). We hope that our updated results showing more robustness to layer and $\epsilon_\text{init}$ inspire confidence that GRACE surpasses existing alternatives, which themselves have even more hyperparameters to tune.
2. *A relaxed, episodic lifelong setting*: In practice, as data stream in and datasets are collected, some models can be retrained or finetuned periodically. But this is expensive and some edits are safety-critical and time-sensitive, so GRACE can edit models between bigger updates. Here, there may also be ways to leverage exogenous or historical data to tune or refresh GRACE codebooks.

In both settings, we believe GRACE poses an exciting, practical solution that surpasses existing works.

**Individual Responses**. We address each reviewer’s comments individually below. We have worked hard to address your concerns and hope you find our responses informative. If you feel our comments have not sufficiently addressed your concerns, we would love to discuss with you further.

---

### Decision · Program_Chairs · 2023-09-21

**Decision:**

Accept (poster)

**Comment:**

This paper has received the scores of 4, 6, 7, 7, 7. Four reviewers voted for accepting the paper, but one reviewer has initially given the score of 4, but the review were rather short, and s/he did not respond to the authors' rebuttal. Given the importance of the problem and the technical contributions, as well as the relatively high review scores from the reviewers, I recommend to accept the paper as a poster.